# WinNet:time series forecasting with a window -enhanced period extracting and Interacting

## Abstract

Recently, Transformer-based methods have significantly improved state-of-the-art time series forecasting results, but they suffer from high computational costs and the inability to capture the long and short periodicity of time series. We present a highly accurate and simply structured CNN-based model for long-term time series forecasting tasks, called WinNet, including (i) Inter-Intra Period Encoder (I2PE) to transform 1D sequence into 2D tensor with long and short periodicity according to the predefined periodic window, (ii) Two-Dimensional Period Decomposition (TDPD) to model period-trend and oscillation terms, and (iii) Decomposition Correlation Block (DCB) to leverage the correlation of the period-trend and oscillation terms to support the prediction tasks by CNNs. Results on nine benchmark datasets show that the WinNet can achieve SOTA performance and lower computational complexity over CNN-, MLP-, Transformer-based approaches. The WinNet provides potential for the CNN-based methods in the time series forecasting tasks, with perfect tradeoff between performance and efficiency.

## 1 INTRODUCTION

Time series forecasting (TSF) has been widely used in the prediction of energy consumption, transportation, economic planning, weather and disease transmission. The TSF tasks are to leverage the known sequence of multiple time steps to predict the information of multiple time steps in the future, which further facilitates resource planning and management. Extensive neural architectures have been designed to achieve the TSF tasks. Recent deep learning models have achieved significant performance improvements, such as Informer (Zhou et al., 2021), AutoFormer (Wu et al., 2021), FEDformer (Zhou et al., 2022), DLinear (Zeng et al., 2023), TimesNet (Wu et al., 2023), PatchTST (Nie et al., 2023). Benefiting from the self-attention mechanism, the Transformer-based models are able to capture the long-term dependency of temporal sequence, achieving state-of-the-art (SOTA) performance for TSF tasks. However, these models are not sensitive to the periodicity and have high computational complexity. Recently, the DLinear outperforms the Transformer-based architectures with only a single linear layer, which results in increasing research attentions and discussions between the MLP-based and the Transformer-based architectures. In the TimesNet, the classical convolutional neural network (CNN) is applied to extract the periodic features after converting the sequence into two-dimensional (2D) tensor by multi-periods, which also inspires us to reconsider the CNN-based methods in the TSF tasks.

Since the future status of a system is time-evolving and with uncertainties, the TSF tasks can be quite challenging. Except for the temporal and regular changes, the uncertainties of the time series data, as well as the noise inputs, provide extra technical difficulties to apply the trend and seasonal terms to achieve the TSF tasks. However, the performance of the model is strongly correlated with the periodicity. A new method of setting up a periodic window is proposed to process the multi-periods of the time series and the corresponding neural network model is designed to capture the complicated underlying patterns of temporal sequence. The network mainly extracts periodicity of the time series through the periodic **win**dow, called WinNet.

In the WinNet, the original sequence is transformed by MLP layer to extract the periodicity. The periodic window is approximated as the least common multiple of multi-periods obtained by the Fast Fourier Transformation (FFT) (Wu et al., 2023). In this way, the periodic window can represent the variation of multiple short periods, and the sequence is organized into 2D tensor according to the

periodic window. In the 2D tensor, each row represents the short-period trend within the periodic window, and each column represents the long-period trend of the whole sequence. Subsequently, the features of long and short periods are separately extracted by the I2PE. The TDPD module is proposed to decompose the 2D tensor into the period-trend and oscillation terms, which highlights the importance of periodicity. Based on the correlation analysis, we find that there are extremely strong lag-correlations between the trend and seasonal terms, and the correlation has a periodic pattern, as shown in Figure 1. To mine their correlation instead of utilizing them separately like DLinear and MICN, the DCB is innovatively designed to combine the period-trend and oscillation terms using a convolutional kernel. The learned weights of the convolution kernel represent the importance of the period-trend and oscillation terms of neighboring time steps. To perform an efficient periodic fusion of the time series, the Series Decoder is proposed to interactively combine the features of long and short period and map the learned features into the prediction of time steps. The WinNet can reduce the relative Mean Squared Error (MSE) and Mean Absolute Error (MAE) in multivariate time series by 18.5% and 12.0%, respectively, compared to TimesNet.

Figure 1: The lag correlations of the trend and seasonal terms in ETTm1 and ECL datasets. We can see that the lag-correlations between the two terms are very strong and there is a periodic pattern.

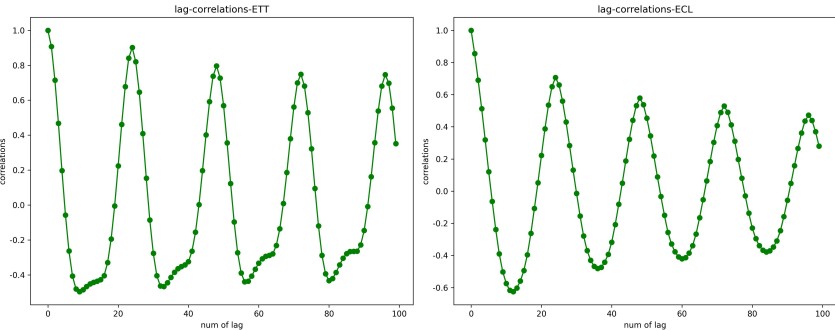

In summary, this work contributes the time series forecasting tasks in the following ways:

- Only one convolutional layer is designed as the backbone of the prediction network, which greatly reduces the training memory and computational complexity and improves experimental efficiency. This also indicates that the simple model architecture can also be effective for the TSF tasks.

- The time series are reorganized according to a periodic window, which can represent the trend variation of multiple short periods.

- To enhance the modeling ability, time series are further decomposed into the period-trend and oscillation terms by the TDPD module. The DCB is proposed to aggregate the neighboring periodic information to obtain the local periodicity by extracting the correlation between the two terms.

- Extensive experiments are conducted on 9 benchmark datasets across multiple domains (energy, traffic, economics, weather, electricity and illness). Our experimental results demonstrate that the WinNet outperforms other comparative baselines in both the univariate and multivariate prediction tasks with long and short input lengths. The WinNet provides potential for the CNN-based methods in the TSF tasks.

## 2 RELATED WORK

It is widely recognized that the uncertainties of the temporal sequence provide extra difficulties in the TSF tasks. In recent years, extensive deep learning models have been proposed to achieve the temporal modeling, including RNN-based, CNN-based, Transformer-based, and MLP-based models. The SOTA performance and specific advantages are demonstrated by extensive experiments, as shown below:

**RNNs** In general, RNN networks are the primary tools for temporal modeling before the Transformer architecture. RNN-based methods, such as LSTM (Hochreiter & Schmidhuber, 1997), GRU (Chung et al., 2014) and DeepAR (Salinas et al., 2020), utilize the recurrent information transmission to capture the temporal changes through state transitions among time steps.

**Transformers** Transformer (Vaswani et al., 2017) and its variants are also initially proposed to achieve the Natural Language Processing tasks, with the advantage of parallel recurrent computation and the self-attention mechanism to capture the long-term temporal correlation. Currently Transformer has been widely used in TSF tasks, such as the Informer, Autoformer, FEDformer, Crossformer (Zhang & Yan, 2023), PatchTST, PETformer (Lin et al., 2023), etc. The attention mechanism is designed to capture the temporal dependencies among time steps and achieve significant performance in the TSF tasks. In the Autoformer, the sequence trend decomposition is to capture the temporal pattern of a sequence by the trend and seasonal terms, and an auto-correlation mechanism is to capture the temporal dependence of the series based on learning periods. In the FEDformer, a Fourier enhanced structure is designed to enhance sparse attention in the frequency domain. By referring to the Vision Transformer (Dosovitskiy et al., 2020), the PatchTST utilizes the slice and dice strategy to formulate the patch and the Transformer architecture is applied to extract local semantic information of the multivariate time series.

**MLPs** DLinear has been proven to be effective for TSF, which achieve competitive performance over the Tranformer architecture by a simple one-layer linear transformation with channel independence (CI). Since then, many MLP-based models are proposed to encode temporal dependencies for the TSF tasks, including LightTS (Zhang et al., 2022), MTS-Mixers (Li et al., 2023b), TSMixer (Ekambaram et al., 2023), RLinear (Li et al., 2023a), etc. The MLP-based methods significantly address the computational efficiency issue of Transformer, and their simplified model structures allow them to be another prominent architecture for TSF tasks.

**CNNs** CNN networks were mainly used in the field of Computer Vision, where they can be used to extract local information from images. Recently, CNNs have been proposed to mine the periodicity of time series, such as TCN (Bai et al., 2018), TimesNet, MICN, etc. The TCN captures temporal changes through a convolutional kernel that slides along the temporal dimension. In the TimesNet, the original sequence is reshaped into the 2D tensor by periods to extract multiple periods of the sequence by FFT. The classical InceptionV1 network (Szegedy et al., 2015) is applied to process the temporal samples within the periods to support the forecasting tasks. A multi-scale isometric convolutional network with multi-scale branches is proposed in the MICN to capture both local and global features from a holistic view of the temporal sequence.

## 3 WINNET

The general architecture of WinNet is illustrated in Figure 2. As mentioned before, based on the multi-periodic features of time series, the periodic window with the superposition of multiple periods is proposed to capture the periodic changes of the time series within the periodic window. I2PE block is designed to extract the periodicity of the sequence and obtain intra-period and its transpose (inter-period). The intra-period indicates that the rows of the 2D tensor are the periodic windows, while the inter-period indicates that the columns are the periodic windows. The TDPD is separately performed on the two features to obtain the period-trend and oscillation terms. The DCB is followed to learn the local correlation between the period-trend and oscillation terms, respectively. The final prediction results are obtained by the Series Decoder.

### 3.1 INTER-INTRA PERIOD ENCODER

In DLinear, a single linear layer has the ability to effectively capture periodicity in the time series. We perform a linear layer on the original sequence (Li et al., 2023b), and the linear mapped sequence can acquire the periodic information from each sample in the original sequence. We find that the MLP layer can reduce the period of the original sequence dramatically, making the period characteristics more obvious and facilitating the period extraction for CNN network. For the top-k periods obtained by FFT in Appendix Table 8, multiple short periods are encapsulated within the periodic window, facilitating the extraction of multiple short periods using the convolutional networks for

Figure 2: The model architecture of WinNet. The period and osc represent the period-trend and oscillation terms. CI and CA mean the channel independence and aggregation strategy, similiar to DLinear(Zeng et al., 2023), and $sl, pl$ indicate the input and prediction length. $c$, $n$, $w$ represent the number of channel, periodic window and the periodic window size, respectively. The output (in blue) is the final result of the TSF tasks.

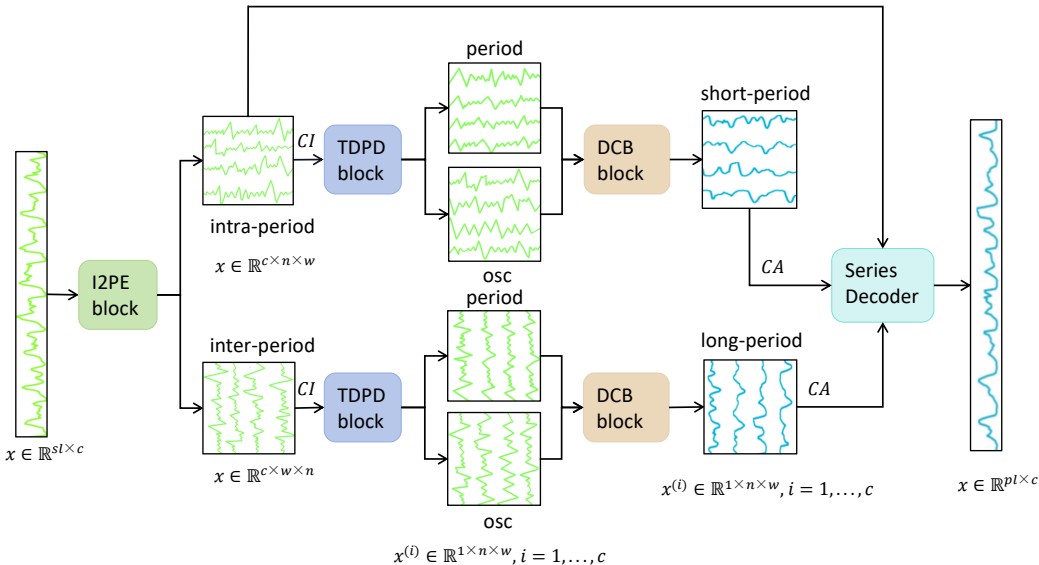

complex sequences. As shown in Appendix Figure 8, the periodic window is approximated as the least common multiple of each refined period.

After the operations in I2PE block, the original sequence is reshaped into a 2D tensor according to the size of the periodic window, as shown in the equation 1:

$$\hat{\mathbf{X}}_{1D} = \text{Permute}(\text{RevIN}(\mathbf{X}_{1D}))$$
$$\mathbf{X}_{2D}^{row} = \text{Reshape}(\text{Linear}(\hat{\mathbf{X}}_{1D})) \quad (1)$$
$$\mathbf{X}_{2D}^{col} = \text{Transpose}(\mathbf{X}_{2D}^{row})$$

where $\mathbf{X}_{1D} \in \mathbb{R}^{sl \times c}$ is the original sequence, $\mathbf{X}_{2D}^{row} \in \mathbb{R}^{c \times n \times w}$ is the intra-period whose rows represent the periodic windows, while $\mathbf{X}_{2D}^{col} \in \mathbb{R}^{c \times w \times n}$ represents its columns are the periodic windows. The $sl$ and $c$ denote the input length and the number of channels in the sequence, and the $n$, $w$ are the number of periodic windows and the periodic window size (in the experiments, $n = w$). RevIN's regularization method is referenced from NLinear (Zeng et al., 2023).

Specifically, each row in the intra-period represents a periodic window with the superposition of multiple short periods, and multiple windows are organized into each column to capture the variation of the time series among periodic windows. Since the periodic window size is approximated as a common multiple of the top-k periods of the sequence, the long-periodicity correlation can be found among the column at the corresponding positions in both periodic windows.

## 3.2 TWO-DIMENSIONAL PERIOD DECOMPOSITION

In general, existing methods for time series trend decomposition mainly focused on trend decomposition of 1D sequence. In this work, inspired by the trend decomposition idea in DLinear, we propose a TDPD strategy, as shown in equation 2. Specifically, a trend-padding operation is dedicatedly designed to perform the convolutional operation at the boundary. As shown in Figure 4, after the operation, the 2D tensor $\mathbf{X}_{2D} \in \mathbb{R}^{s \times n \times w}$ are padded into a new 2D tensor $\mathbf{X}_{2D} \in \mathbb{R}^{s \times (n+p) \times (w+p)}$ where $p$ means the padding lengths in row or column.

Figure 3: The figure of TrendPadding. Unlike the 0 or same padding mode in common CNNs, the neighbor samples (before or after) in the original sequence are selected as the padding item to retain the trend characteristics of the whole sequence. To keep the shape of the matrix, we complement the remaining positions with 0.

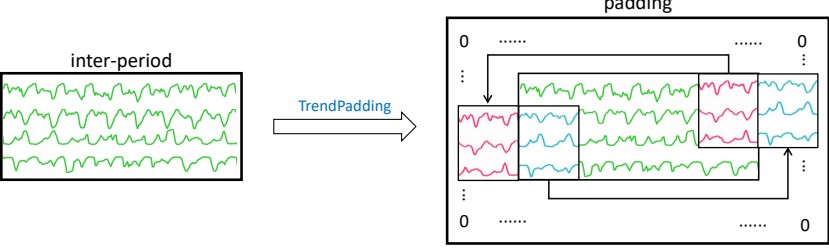

$$\mathbf{X}_{period} = \text{AvgPool2D}(\text{TrendPadding}(\mathbf{X}_{2D}))_{k \times k}$$
$$\mathbf{X}_{osc} = \mathbf{X} - \mathbf{X}_{period}$$

(2)

where $\mathbf{X}_{period}$ and $\mathbf{X}_{osc}$ indicate the period-trend and oscillation terms, respectively, and $k \times k$ is the size of the $\text{Avgpool2D}(\cdot)$ kernel size.

Common $\text{Avgpool1D}(\cdot)$ operation focuses on the average change in the trend of time series, while $\text{Avgpool2D}(\cdot)$ is applied to extract both the trend within the periodic window (intra-correlation) and the changes of the long period among the neighbouring windows (inter-correlation). According to the equation 2, the two features can be decomposed into the period-trend and oscillation terms. Specifically, the period-trend term keeps a balance between intra-correlation trends and inter-correlation periodicity. As shown in Appendix Figure 7, we can see that the trend of the time steps among each window keeps essentially the same.

### 3.3 DECOMPOSITION CORRELATION BLOCK

After decomposing the sequence into the trend and seasonal terms, DLinear simply feeds the trend and seasonal terms independently into a linear layer for model training. MICN predicts the seasonal term by the proposed MIC layer, while the trend-cyclical part is directly obtained by linear regression. They fail to capture the correlation between trend and seasonal terms.

Figure 4: The figure of DCB. The period and osc represent the period-trend and oscillation terms. Use a convolutional kernel to extract the variation of the two terms within the periodic neighborhood.

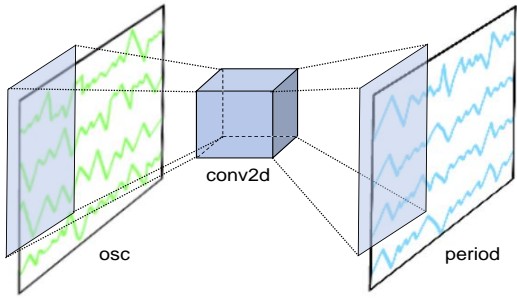

We believe that there is a correlation between period-trend and oscillation terms and that they are influencing future time steps together. Specifically, time steps in the sequence can be affected by the period-trend and oscillation terms obtained by TDPD within the moments of periodic neighborhood. The CNN kernel can exactly extract the variation of the two terms within periodic neighborhood, and the learned parameters are able to perform a proportional aggregation of them, instead of simply adding. We choose CNN as our backbone network to synthesize the sequence information of $N$

time steps in the periodic neighborhood. The process is described below:

$$\mathbf{X}_{period}^{CI}, \mathbf{X}_{osc}^{CI} = \text{CI}(\mathbf{X}_{period}), \text{CI}(\mathbf{X}_{osc})$$
$$\mathbf{X}_{input}^{CI} = \text{Concat}(\mathbf{X}_{period}^{CI}, \mathbf{X}_{osc}^{CI})$$
$$\hat{\mathbf{X}}_{output}^{CI} = \text{Dropout}(\text{Sigmoid}(\text{Conv2D}(\mathbf{X}_{input}^{CI})))$$
$$\hat{\mathbf{X}}_{output}^{CI} = \text{CA}(\hat{\mathbf{X}}_{output}^{CI})$$

(3)

where $\text{CI}(\cdot)$ and $\text{CA}(\cdot)$ mean channel independence and aggregation strategy. In the $\text{CI}(\cdot)$, we split both period-trend and oscillation terms at the channel dimension and concatenate them into two-channel matrices. We feed the matrices at the same channel into the DCB block. In the $\text{CA}(\cdot)$, we concatenate the single-channel output after the DCB block at the channel dimension for fusion.

### 3.4 SERIES DECODER

The DCB can learn the local correlative features of the time series, and in this section, the Series Decoder is designed to aggregate the inter-period and intra-period for extracting global periodicity of each window. Specifically, the process is:

$$\hat{\mathbf{X}}_{i,j}^{fusion} = \hat{\mathbf{X}}_{i,j}^{row} + \hat{\mathbf{X}}_{j,i}^{col}, i,j \in (1,2,...,w)$$
$$\hat{\mathbf{X}}_{i,j}^{res} = \hat{\mathbf{X}}_{i,j} + \mathbf{X}_{i,j}^{row}$$
$$\hat{\mathbf{X}}^{final} = \text{Permute}(\text{Linear}(\text{Reshape}(\hat{\mathbf{X}}_{2D}^{res})))$$

(4)

where $i,j$ together denote a temporal point of the 2D tensor, $w$ represents the size of the periodic window, matrix $\hat{\mathbf{X}}_{i,j}^{res}$ and $\hat{\mathbf{X}}_{i,j}^{final}$ are the output of residual connection and the final result.

$\mathbf{X}_{i,j}^{row}$ is obtained by intra-period convolution, reflecting the short periodicity of the sequence, and $\mathbf{X}_{j,i}^{col}$ is obtained by inter-period convolution, which is spaced by window size and reflects the long periodicity of the sequence. This design is able to interactively learn the correlation of the short-period time steps with the fusion of the long-period features, thus further extracting the global periodicity of the sequence. After the operation, a simple linear layer is designed to map the learned features into the prediction of time steps.

## 4 EXPERIMENT

**Datasets** In this section, a total of 9 real-world datasets[1] are applied to validate the proposed approachs and selective baselines, including weather, traffic, ECL, ILI, exchange and ETT datasets (ETTh1, ETTh2, ETTm1, ETTm2). It should be noted that ETTh1, ETTh2 and ILI are small datasets with small channels, ETTm1, ETTm1 and weather are medium datasets with small channels, and ECL, traffic are large datasets with multi-channels. In general, periodicity is more easily captured for small datasets and more difficult for large datasets.

**Baselines and metrics** The following methods are selected as the baselines, including Transformer-based PatchTST, Crossformer, FEDformer, Autoformer, CNN-based TimesNet and MICN, and MLP-based DLinear, RLinear and RMLP. All models follow the same experimental setup with a prediction length of T $\in$ { 24, 36, 48, 60} for the ILI dataset and T $\in$ { 96, 192, 336, 720} for the other datasets. We collect baseline results from DLinear, PatchTST and TimesNet. The default input length L=96 is for the Transformer-based model and L=512 is for PatchTST/64. To ensure the effectiveness of the Transformer and DLinear-based methods, two input lengths {96 and 512}, are applied to conduct the performance comparison with SOTA models. In addition, we also explore the influence of the size of the input length on the performance of the proposed model. The Mean Squared Error (MSE) and Mean Absolute Error (MAE) are selected to measure the model performance for both the multivariate and univariate TSF tasks. Note that a smaller value indicates higher performance for both the MSE and MAE. In the following experimental results, the best results are marked in red and the next best in blue. Avg is the averaged result from all four prediction lengths. All experiments are implemented in PyTorch and conducted on a single NVIDIA RTX3090 24GB GPU.

---

[1] https://drive.google.com/drive/folders/13Cg1KYOlzM5C7K8gK8NfC-F3EYxkM3D2

## 5 EXPERIMENTAL RESULTS

The results for multivariate and univariate predictions on time series datasets are summarized in Tables 1, 2 and 3. In general, our model outperforms the selected baselines on both multivariate and univariate forecasting tasks. Detailed results of the experiment can be found in the Appendix A.4.

**Multivariate Results** For multivariate sequence prediction, as shown in Tables 1 and 2, in general, the WinNet essentially achieves the best performance for the listed datasets on the two measurements. Quantitatively, from the results of the long-input length experiments, WinNet improves 18.5% in MSE and 12.0% in MAE compared to the CNN-based SOTA model TimesNet, indicating that WinNet can more stably capture the long and short periodicity in data. As for the results of the short-input experiments, compared to the CNN-based TimesNet, the WinNet improves 9.3% in MSE and 8.4% in MAE; compared to the MLP-based DLinear, the WinNet improves 11.9% in MSE and 9.6% in MAE. It is noted that our model achieves a complete outperformance on the ETT datasets for both long and short sequences.

**Univariate Results** The results of univariate prediction is shown in Table 3. The WinNet significantly outperforms other SOTA models on all datasets, achieving a improvement of 8.2% in MSE and 5.0% in MAE for PatchTST, 12.3% in MSE and 8.1% in MAE for TimesNet, 18.9% in MSE and 13.1% in MAE for DLinear. This demonstrates that the modules in WinNet indeed bring more useful periodic information for univariate TSF tasks.

Table 1: Results for multivariate long-input length prediction. The input sequence length is set to 104 for the ILI dataset and 512 for the others. See Appendix Table 16 for the full results.

| Methods | WinNet (Ours) | | RLinear (2023a) | | RMLP (2023a) | | PatchTST (2023) | | TimesNet (2023) | | MICN (2023) | | Crossformer (2023) | | DLinear (2023) | | FEDformer (2022) | | Autoformer (2021) | |
|---|---|---|---|---|---|---|---|---|---|---|---|---|---|---|---|---|---|---|---|---|---|
| Metric | MSE | MAE | MSE | MAE | MSE | MAE | MSE | MAE | MSE | MAE | MSE | MAE | MSE | MAE | MSE | MAE | MSE | MAE | MSE | MAE |
| ETTm1 | **0.345** | **0.371** | 0.378 | 0.401 | 0.367 | 0.397 | 0.352 | 0.382 | 0.407 | 0.417 | 0.372 | 0.399 | 0.424 | 0.438 | 0.352 | 0.381 | 0.382 | 0.422 | 0.515 | 0.493 |
| ETTm2 | **0.248** | **0.310** | 0.281 | 0.346 | 0.291 | 0.350 | 0.256 | 0.316 | 0.283 | 0.336 | 0.299 | 0.357 | 0.418 | 0.432 | 0.267 | 0.331 | 0.291 | 0.343 | 0.310 | 0.356 |
| ETTh1 | **0.402** | **0.419** | 0.442 | 0.456 | 0.461 | 0.468 | 0.418 | 0.432 | 0.485 | 0.481 | 0.523 | 0.513 | 0.440 | 0.454 | 0.424 | 0.438 | 0.428 | 0.453 | 0.473 | 0.476 |
| ETTh2 | **0.332** | **0.385** | 0.469 | 0.463 | 0.425 | 0.448 | 0.342 | **0.385** | 0.414 | 0.445 | 0.668 | 0.557 | 0.443 | 0.455 | 0.431 | 0.446 | 0.387 | 0.434 | 0.422 | 0.442 |
| ILI | 1.919 | 0.912 | 2.347 | 1.101 | 2.350 | 1.084 | **1.538** | **0.841** | 2.345 | 1.037 | 2.526 | 1.060 | 3.394 | 1.215 | 2.169 | 1.041 | 2.596 | 1.069 | 2.819 | 1.119 |
| Exchange | 0.401 | **0.415** | 0.466 | 0.451 | 0.495 | 0.515 | **0.392** | 0.416 | 0.618 | 0.557 | 0.440 | 0.474 | 0.798 | 0.693 | 0.416 | 0.430 | 0.477 | 0.477 | 0.613 | 0.539 |
| Weather | **0.219** | **0.263** | 0.231 | 0.294 | 0.231 | 0.278 | 0.229 | 0.265 | 0.252 | 0.287 | 0.245 | 0.300 | 0.227 | 0.285 | 0.231 | 0.280 | 0.310 | 0.357 | 0.335 | 0.379 |
| Traffic | 0.417 | 0.285 | 0.419 | 0.290 | 0.404 | 0.280 | **0.396** | **0.265** | 0.616 | 0.334 | 0.489 | 0.300 | 0.528 | 0.293 | 0.433 | 0.295 | 0.603 | 0.372 | 0.616 | 0.383 |
| Electricity | **0.159** | **0.253** | 0.167 | 0.261 | 0.162 | 0.256 | 0.161 | **0.253** | 0.200 | 0.301 | 0.188 | 0.296 | 0.304 | 0.355 | 0.166 | 0.263 | 0.207 | 0.321 | 0.214 | 0.326 |

Table 2: Results for multivariate short-input length prediction. The input sequence length is set to 36 for the ILI dataset and 96 for the others. See Appendix Table 17 for the full results.

| Methods | WinNet (Ours) | | RLinear (2023a) | | RMLP (2023a) | | PatchTST (2023) | | TimesNet (2023) | | MICN (2023) | | Crossformer (2023) | | DLinear (2023) | | FEDformer (2022) | | Autoformer (2021) | |
|---|---|---|---|---|---|---|---|---|---|---|---|---|---|---|---|---|---|---|---|---|---|
| Metric | MSE | MAE | MSE | MAE | MSE | MAE | MSE | MAE | MSE | MAE | MSE | MAE | MSE | MAE | MSE | MAE | MSE | MAE | MSE | MAE |
| ETTm1 | **0.381** | **0.385** | 0.395 | 0.404 | 0.400 | 0.414 | 0.382 | 0.395 | 0.392 | 0.413 | 0.400 | 0.405 | 0.470 | 0.468 | 0.403 | 0.406 | 0.448 | 0.452 | 0.587 | 0.517 |
| ETTm2 | **0.276** | **0.320** | 0.312 | 0.366 | 0.330 | 0.370 | 0.285 | 0.330 | 0.328 | 0.382 | 0.291 | 0.332 | 0.470 | 0.468 | 0.350 | 0.400 | 0.304 | 0.349 | 0.327 | 0.370 |
| ETTh1 | **0.439** | **0.425** | 0.466 | 0.462 | 0.462 | 0.454 | 0.470 | 0.452 | 0.558 | 0.535 | 0.460 | 0.455 | 0.518 | 0.503 | 0.455 | 0.451 | 0.440 | 0.459 | 0.496 | 0.487 |
| ETTh2 | **0.375** | **0.400** | 0.460 | 0.459 | 0.515 | 0.482 | 0.384 | 0.406 | 0.587 | 0.525 | 0.414 | 0.427 | 0.491 | 0.486 | 0.558 | 0.515 | 0.436 | 0.449 | 0.449 | 0.459 |
| ILI | 2.388 | 0.977 | 3.061 | 1.202 | 3.483 | 1.280 | **1.833** | **0.845** | 2.664 | 1.085 | 2.138 | 0.930 | 4.251 | 1.406 | 2.615 | 1.090 | 2.846 | 1.143 | 3.006 | 1.161 |
| Exchange | 0.373 | 0.406 | 0.339 | 0.401 | 0.396 | 0.432 | 0.367 | **0.402** | **0.334** | 0.425 | 0.416 | 0.443 | 0.764 | 0.651 | 0.354 | 0.413 | 0.518 | 0.499 | 0.613 | 0.539 |
| Weather | 0.250 | 0.293 | 0.267 | 0.320 | 0.251 | 0.295 | 0.257 | **0.279** | 0.242 | 0.299 | 0.259 | 0.286 | 0.248 | 0.305 | 0.265 | 0.316 | 0.308 | 0.360 | 0.337 | 0.381 |
| Traffic | **0.457** | 0.356 | 0.630 | 0.390 | 0.546 | 0.352 | 0.541 | 0.348 | 0.541 | 0.315 | 0.619 | 0.335 | 0.558 | **0.309** | 0.624 | 0.383 | 0.609 | 0.376 | 0.627 | 0.378 |
| Electricity | 0.192 | **0.280** | 0.206 | 0.298 | 0.202 | 0.291 | 0.211 | 0.297 | **0.186** | 0.294 | 0.192 | 0.295 | 0.320 | 0.372 | 0.211 | 0.300 | 0.213 | 0.326 | 0.227 | 0.337 |

Table 3: Results for univariate long-input length prediction. The input sequence length is set to 104 for the ILI dataset and 336 for the others. See Appendix Table 18 for the full results.

| Methods | WinNet (Ours) | | RLinear (2023a) | | RMLP (2023a) | | PatchTST (2023) | | TimesNet (2023) | | MICN (2023) | | DLinear (2023) | | FEDformer (2022) | | Autoformer (2021) | |
|---|---|---|---|---|---|---|---|---|---|---|---|---|---|---|---|---|---|---|
| Metric | MSE | MAE | MSE | MAE | MSE | MAE | MSE | MAE | MSE | MAE | MSE | MAE | MSE | MAE | MSE | MAE | MSE | MAE |
| ETTm1 | **0.044** | **0.157** | 0.054 | 0.170 | 0.074 | 0.208 | 0.048 | 0.162 | 0.053 | 0.173 | 0.049 | 0.164 | 0.053 | 0.167 | 0.069 | 0.201 | 0.080 | 0.221 |
| ETTm2 | **0.109** | **0.246** | 0.114 | 0.253 | 0.133 | 0.276 | 0.112 | 0.251 | 0.132 | 0.275 | 0.111 | 0.247 | 0.112 | 0.247 | 0.119 | 0.261 | 0.129 | 0.271 |
| ETTh1 | **0.069** | **0.207** | 0.105 | 0.249 | 0.123 | 0.277 | 0.073 | 0.210 | 0.074 | 0.215 | 0.102 | 0.251 | 0.103 | 0.246 | 0.111 | 0.257 | 0.104 | 0.252 |
| ETTh2 | 0.178 | **0.334** | 0.205 | 0.359 | 0.222 | 0.374 | **0.176** | 0.336 | 0.180 | 0.341 | 0.190 | 0.342 | 0.198 | 0.350 | 0.205 | 0.349 | 0.217 | 0.363 |
| Weather | **0.0013** | **0.0280** | 0.0063 | 0.0662 | 0.0041 | 0.0498 | 0.0014 | 0.0282 | 0.0015 | 0.0295 | 0.0064 | 0.0675 | 0.0062 | 0.0665 | 0.0042 | 0.0526 | 0.0063 | 0.0581 |
| Exchange | 0.484 | **0.472** | 0.520 | 0.519 | 0.574 | 0.583 | **0.456** | 0.508 | 0.583 | 0.535 | 0.534 | 0.543 | 0.566 | 0.544 | 0.725 | 0.637 | 0.789 | 0.681 |
| ECL | 0.258 | **0.359** | **0.257** | **0.359** | 0.284 | 0.381 | 0.400 | 0.442 | 0.307 | 0.394 | 0.317 | 0.412 | **0.257** | 0.360 | 0.456 | 0.507 | 0.551 | 0.558 |
| Traffic | **0.132** | **0.217** | 0.134 | 0.219 | 0.148 | 0.238 | 0.141 | 0.223 | 0.144 | 0.234 | 0.152 | 0.240 | 0.144 | 0.238 | 0.302 | 0.398 | 0.263 | 0.370 |
| ILI | **0.665** | **0.613** | 1.921 | 1.223 | 1.095 | 0.920 | 0.794 | 0.684 | 0.777 | 0.723 | 1.279 | 0.913 | 0.714 | 0.695 | 1.107 | 0.922 | 1.139 | 0.931 |

Table 4: Results for ablation studies, including the I2PE, TDPD and DCB in WinNet. Four cases are included: (a) all the three modules are included in model (Final: I2PE+TDPD+DCB); (b) only the TDPD; (c) TDPD+DCB; (d) the original version with common CNN and one-dimensional trend decomposition.

| Methods | | WinNet | | | | | | | | TimesNet∗ | | DLinear | |
| | | Final | | TDPD+DCB | | TDPD | | original | | | | | |
| Metric | | MSE | MAE | MSE | MAE | MSE | MAE | MSE | MAE | MSE | MAE | MSE | MAE |
|---|---|---|---|---|---|---|---|---|---|---|---|---|---|
| Weather | 96 | **0.143** | **0.198** | 0.147 | 0.203 | 0.147 | 0.206 | 0.147 | 0.208 | 0.163 | 0.223 | 0.176 | 0.237 |
| | 192 | **0.188** | **0.240** | 0.191 | 0.244 | 0.191 | 0.244 | 0.193 | 0.253 | 0.218 | 0.266 | 0.192 | 0.246 |
| | 336 | **0.235** | **0.280** | 0.240 | 0.290 | 0.245 | 0.290 | 0.246 | 0.300 | 0.280 | 0.306 | 0.240 | 0.287 |
| | 720 | **0.310** | **0.336** | 0.315 | 0.340 | 0.321 | 0.342 | 0.326 | 0.357 | 0.349 | 0.356 | 0.316 | 0.352 |
| ECL | 96 | **0.129** | **0.225** | 0.135 | 0.231 | 0.139 | 0.237 | 0.145 | 0.249 | 0.181 | 0.281 | 0.140 | 0.237 |
| | 192 | **0.147** | **0.240** | 0.150 | 0.244 | 0.153 | 0.249 | 0.163 | 0.266 | 0.193 | 0.293 | 0.153 | 0.249 |
| | 336 | **0.163** | **0.257** | 0.167 | 0.261 | 0.169 | 0.264 | 0.180 | 0.283 | 0.205 | 0.312 | 0.169 | 0.267 |
| | 720 | **0.198** | **0.290** | 0.211 | 0.300 | 0.208 | 0.297 | 0.217 | 0.315 | 0.222 | 0.320 | 0.203 | 0.301 |
| traffic | 96 | **0.394** | **0.274** | 0.405 | 0.281 | 0.414 | 0.288 | 0.528 | 0.300 | 0.603 | 0.328 | 0.410 | 0.282 |
| | 192 | **0.407** | **0.279** | 0.420 | 0.287 | 0.428 | 0.294 | 0.549 | 0.312 | 0.610 | 0.329 | 0.423 | 0.287 |
| | 336 | **0.416** | **0.283** | 0.437 | 0.295 | 0.442 | 0.302 | 0.569 | 0.323 | 0.619 | 0.330 | 0.436 | 0.296 |
| | 720 | **0.453** | **0.305** | 0.465 | 0.312 | 0.468 | 0.316 | 0.610 | 0.340 | 0.632 | 0.352 | 0.466 | 0.315 |

∗ We replace the input length L=512 in TimesNet for a fair comparison.

# 6 ABLATION STUDIES

**Model architecture** To validate the proposed modules in the WinNet, the ablation studies are conducted to determine the best model architecture, including the I2PE, TDPD, and DCB. The TimesNet and DLinear are as a SOTA benchmark for CNN-based and MLP-based models. Based on the results in Table 4, we can see that all the proposed modules can significantly improve the prediction performance, which validates the proposed model architecture. We also explore the effect of both the intra-period and inter-period on the proposed model performance, as shown in Appendix Table 11 and 12. Ablation study results on ETT datasets are available in Appendix Table 13.

In the original version, normal trend decomposition and a regular CNN network are applied to replace the I2PE and TDPD, respectively. Compared to using the TDPD, the original version fails to capture the periodicity in complex datasets and suffers from inferior results over the DLinear. Taking the traffic dataset as an example, the TDPD module can improve on the MSE by 22.3%, and achieve comparable results with the DLinear. Other modules can also contribute expected performance improvements and finally outperform the SOTA baselines (TimesNet and DLinear).

**Input length** In general, we consider that a larger look-back window can capture a long-range of periodicity, such as 96 for DLinear and 512 for PatchTST. To validate this issue, we compare the model performance with input lengths of 96 and 512, and the results are reported in the Table 1

and 2. In addition, other configurations are also considered in the proposed model to validate the performance, including {24, 48, 96, 192, 336, 512, 720} as the input length and {96, 720} as the prediction length respectively. As can be seen from Figure 5, the predictions of our model are more significant as the prediction length increases.

Figure 5: Prediction error (MSE) with different look-back windows on 3 large datasets: weather, ECL, traffic. The look-back windows are selected to be L = {24, 48, 96, 192, 336, 512, 720}, and the prediction lengths are T = {96, 720}.

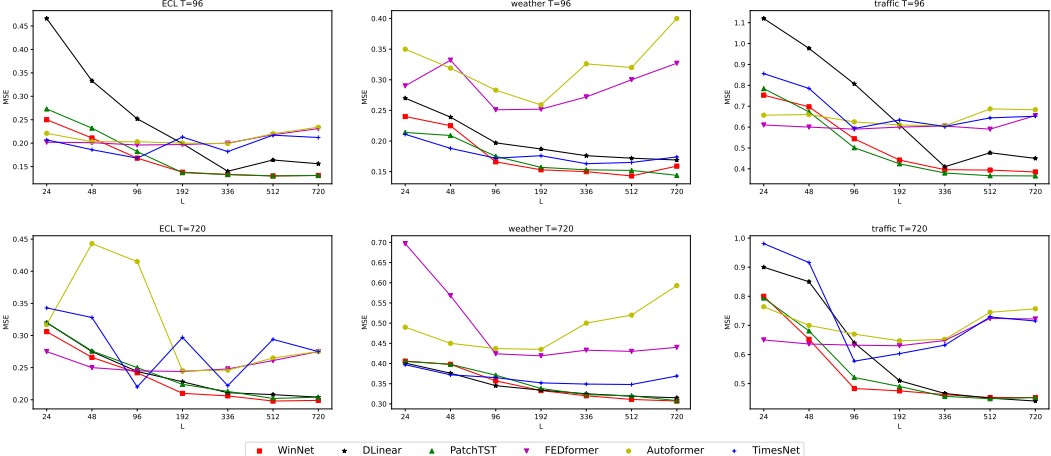

**Model efficiency**    In addition to the expected performance improvements, we also harvest a higher computational efficiency. Table 5 shows the computational efficiency of our model in the univariate prediction tasks. From the table, we can see that our model can achieve higher efficiency, in terms of computational complexity, number of parameters and memory consumption, even over the simple DLinear model. The efficiency of WinNet on the multi-channel and few-channel datasets can be seen in the Appendix Table 14 and 15.

Table 5: Efficiency of our model on the Traffic dataset vs. other methods in univariate prediction. We set the input length to 720 and the prediction length to 720. See relevant computational efficiency with thop, torchsummary and torch.cuda.memory_allocated functions. Times-T denotes the time of an iter training, and Times-I denotes the actual inference time.

| Method | WinNet | PatchTST | TimesNet | MICN | Crossformer | DLinear | FEDformer | Autoformer | Informer | Transformer |
|--------|--------|----------|----------|------|-------------|---------|-----------|------------|----------|-------------|
| FLOPs | **851.3K** | 44.2M | 3240.7G | 5.32G | 726.5M | 1.04M | 1.74G | 1.74G | 1.41G | 1.74G |
| Params | **830.8K** | 8.69M | 450.9M | 18.75M | 11.09M | 1.04M | 3.94M | 2.37M | 2.77M | 2.38M |
| Times-T | 17ms | 24ms | 491ms | 25ms | 55ms | **12ms** | 430ms | 265ms | 142ms | 45ms |
| Memory | **11MiB** | 44MiB | 1762MiB | 85MiB | 56MiB | 12MiB | 24MiB | 27MiB | 29MiB | 27MiB |
| Times-I | 9.6ms | 11.2ms | 66.9ms | 12.7ms | 21.2ms | **8.6ms** | 55.0ms | 28.2ms | 18.1ms | 14.1ms |

# 7    CONCLUSIONS

In summary, we propose a CNN-based approach for time series forecasting models by introducing the important modules of periodic window, including the I2PE, TDPD, and DCB. Compared to previous SOTA models, our model captures the correlation between long and short periods and is more effective as the look-back window gets longer. The proposed model not only outperforms other baselines in terms of prediction accuracy, but also harvests higher computational efficiency.

This work demonstrates the potential for the CNN-based methods in the TSF tasks. The correlation between period-trend and oscillation terms can provide the local periodicity in time series. In the future, we should focus on the correlation and interplay between the period-trend and oscillation terms, instead of training them separately.

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

# A  APPENDIX

## A.1  EXPERIMENT DETAILS

**Datasets characteristics**  The dataset is characterised as follows in Appendix Table 6. We follow standard protocol (Nie et al., 2023) and split all datasets into training, validation and test set in chronological order by the ratio of 6:2:2 for the ETT dataset and 7:1:2 for the other datasets. Frequency indicates the sampling time difference between neighbouring time steps.

Table 6: Summary of experiment datasets.

| Datasets | ETTm1 | ETTm2 | ETTh1 | ETTh2 | weather | ILI | exchange | traffic | ECL |
|---|---|---|---|---|---|---|---|---|---|
| Channels | 7 | 7 | 7 | 7 | 21 | 7 | 7 | 862 | 321 |
| Lengths | 69680 | 69680 | 17420 | 17420 | 52696 | 966 | 7588 | 17544 | 26304 |
| Frequency | 15min | 15min | 1h | 1h | 10min | 7day | 1day | 1h | 1h |

Figure 6: The features after the Decomposition Correlation Block. In the figure, the rows represent periodic windows and the columns represent the number of windows.

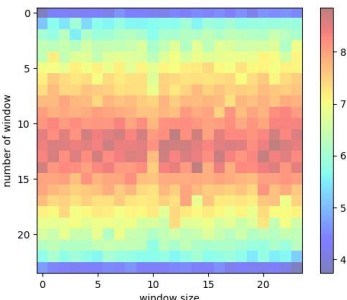

**Decomposition Correlation Block**  Convolutional networks can extract the local correlation between period-trend and oscillation terms based on the weights of the convolutional kernel. A higher weight means the relatively higher correlation. As shown in Appendix Figure 6, we visualise the heat map after the convolution kernel and can see that the model focuses more on the local periodicity of the intermediate window from 5 to 18 for the 2D tensor. In terms of periods, our model pays more attention to the previous 10-15 periods when forecasting.

**Two-Dimensional Period Decomposition**  We take real data from ETTh1 for TDPD and get the period-trend and oscillation terms. As demonstrated in Appendix Figure 7, each column of the period-trend keeps essentially the same trend. In contrast, the oscillation term is characterised by disorganized distributions.

Figure 7: Showcase of the TDPD.

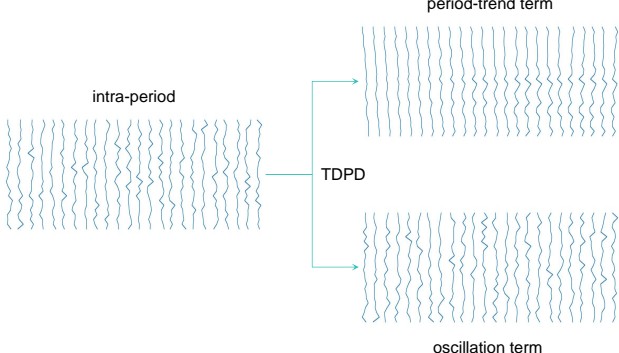

Figure 8: Period Window. The samples with same colour in each row represent time steps in a same period. A sequence can obtain its top-k period values by FFT. The 1st period indicates the most dominant period in the sequence, and so on. For example, the top-3 periods are {6, 4, 3} and approximated multiples of these periods are selected as the size of the periodic window, i.e. 12.

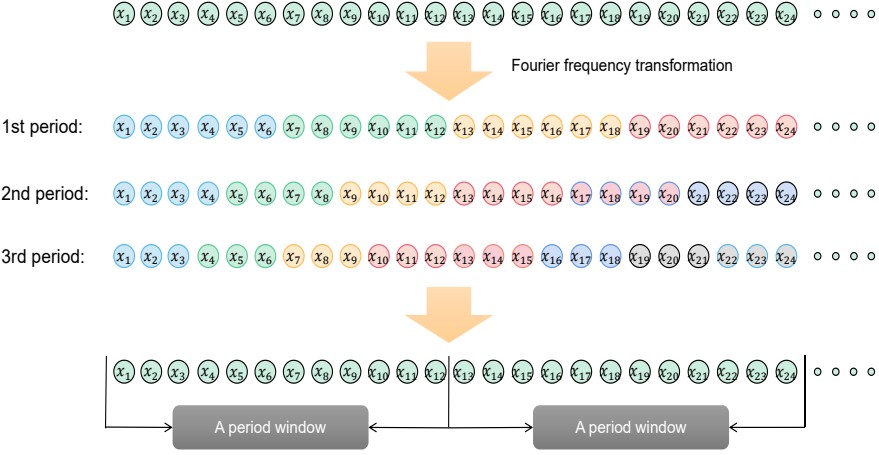

## A.2 MORE ABLATION STUDIES

**Window size** As shown in Appendix Figure 8, since the window size is approximated as a multiple of the multi-short periods, it can also have implications for experimental performance of the proposed model. For example, for a sequence with top-4 periods {4, 6, 8, 12}, we can take 24 as the window size, but it is equally possible to approximate the period of 8 as 9 and take 36 as the final window size. The top-4 periods for the datasets can be seen in Table 8. As demonstrated in Appendix Table 9, the proposed approach can achieve better performance for the periodic window of 24.

Table 7: Period frequencies obtained by FFT for each dataset before MLP. The 1st period represents the period with the highest number of occurrences, and so on. Period represents the specific value of periods and count represents the number of occurrences. It can be seen that most datasets have large periods, difficult to capture the periodicity through CNNs.

| topk periods | 1st period | | 2nd period | | 3rd period | | 4th period | |
|---|---|---|---|---|---|---|---|---|
| statistics | period | count | period | count | period | count | period | count |
| weather | 168 | 759 | 336 | 759 | 112 | 759 | 67 | 759 |
| ETTm1 | 112 | 711 | 84 | 711 | 336 | 711 | 168 | 711 |
| ECL | 168 | 374 | 336 | 294 | 84 | 235 | 112 | 234 |
| ETTm2 | 336 | 711 | 112 | 711 | 84 | 711 | 168 | 711 |
| ETTh1 | 24 | 171 | 336 | 171 | 12 | 171 | 168 | 171 |
| traffic | 24 | 246 | 12 | 246 | 8 | 246 | 168 | 246 |
| exchange | 336 | 101 | 168 | 101 | 112 | 101 | 84 | 101 |
| ETTh2 | 336 | 171 | 24 | 171 | 168 | 171 | 112 | 171 |

Table 8: Period frequencies obtained by FFT for each dataset after MLP. The 1st period represents the period with the highest number of occurrences, and so on. Period represents the specific value of periods and count represents the number of occurrences. 11/12 or 23/24 mean that we use these two periods approximately as one period. Based on these periods, the multiples of them are selected to be the size of the periodic window.

| topk periods | 1st period | | 2nd period | | 3rd period | | 4th period | |
|---|---|---|---|---|---|---|---|---|
| statistics | period | count | period | count | period | count | period | count |
| weather | 3 | 1075 | 2 | 678 | 23/24 | 592 | 11/12 | 234 |
| ETTm1 | 2 | 2528 | 3 | 469 | 23/24 | 355 | 4 | 104 |
| ECL | 2 | 592 | 17 | 372 | 3 | 315 | 8 | 191 |
| ETTm2 | 23/24 | 1185 | 12 | 749 | 288 | 631 | 192 | 351 |
| ETTh1 | 2 | 332 | 3 | 248 | 5 | 98 | 4 | 55 |
| traffic | 2 | 334 | 4 | 328 | 8 | 222 | 3 | 199 |
| exchange | 3 | 125 | 2 | 89 | 6 | 55 | 4 | 48 |
| ETTh2 | 2 | 369 | 3 | 202 | 4 | 108 | 11/12 | 47 |

Table 9: Ablation study with different sizes of the periodic window.

| Methods | Metric | WinNet 18 | | WinNet 24 | | WinNet 32 | | TimesNet | | DLinear | |
|---|---|---|---|---|---|---|---|---|---|---|---|
| | | MSE | MAE | MSE | MAE | MSE | MAE | MSE | MAE | MSE | MAE |
| ETTm1 | 96 | 0.282 | 0.336 | 0.283 | 0.335 | **0.279** | **0.334** | 0.377 | 0.397 | 0.290 | 0.342 |
| | 192 | **0.319** | **0.357** | 0.324 | 0.360 | 0.323 | 0.361 | 0.389 | 0.401 | 0.332 | 0.369 |
| | 336 | **0.354** | **0.379** | 0.357 | **0.379** | **0.354** | 0.382 | 0.393 | 0.414 | 0.366 | 0.392 |
| | 720 | 0.419 | 0.412 | 0.416 | **0.411** | **0.408** | 0.414 | 0.470 | 0.458 | 0.420 | 0.424 |
| ETTm2 | 96 | 0.163 | 0.252 | **0.160** | **0.251** | 0.162 | 0.251 | 0.201 | 0.280 | 0.167 | 0.260 |
| | 192 | 0.216 | 0.290 | **0.212** | **0.287** | 0.216 | 0.291 | 0.242 | 0.313 | 0.224 | 0.303 |
| | 336 | 0.271 | 0.325 | **0.261** | **0.322** | 0.268 | 0.323 | 0.310 | 0.356 | 0.281 | 0.342 |
| | 720 | 0.363 | 0.385 | **0.359** | **0.381** | 0.360 | 0.386 | 0.381 | 0.396 | 0.397 | 0.421 |
| ETTh1 | 96 | 0.367 | 0.392 | **0.362** | **0.390** | 0.374 | 0.397 | 0.460 | 0.464 | 0.375 | 0.399 |
| | 192 | 0.402 | 0.413 | **0.394** | **0.410** | 0.409 | 0.420 | 0.458 | 0.462 | 0.412 | 0.420 |
| | 336 | 0.426 | 0.428 | **0.419** | **0.426** | 0.427 | 0.430 | 0.523 | 0.501 | 0.439 | 0.443 |
| | 720 | 0.442 | 0.454 | 0.436 | 0.453 | **0.434** | **0.451** | 0.502 | 0.497 | 0.472 | 0.490 |
| ETTh2 | 96 | 0.271 | 0.333 | **0.267** | **0.332** | 0.275 | 0.337 | 0.338 | 0.397 | 0.289 | 0.353 |
| | 192 | 0.326 | 0.374 | **0.322** | **0.372** | 0.337 | 0.377 | 0.422 | 0.446 | 0.383 | 0.418 |
| | 336 | 0.356 | **0.400** | **0.351** | 0.401 | 0.368 | 0.414 | 0.431 | 0.460 | 0.448 | 0.465 |
| | 720 | 0.395 | 0.437 | **0.389** | **0.436** | 0.406 | 0.448 | 0.467 | 0.480 | 0.605 | 0.551 |
| weather | 96 | 0.151 | 0.207 | **0.143** | **0.198** | 0.146 | 0.202 | 0.163 | 0.223 | 0.176 | 0.237 |
| | 192 | 0.196 | 0.253 | **0.188** | **0.240** | 0.191 | 0.245 | 0.218 | 0.266 | 0.192 | 0.246 |
| | 336 | 0.245 | 0.288 | **0.235** | **0.280** | 0.239 | 0.289 | 0.280 | 0.306 | 0.240 | 0.287 |
| | 720 | 0.319 | 0.341 | **0.310** | **0.336** | 0.315 | 0.345 | 0.349 | 0.356 | 0.316 | 0.352 |
| ECL | 96 | 0.141 | 0.238 | **0.130** | **0.226** | 0.142 | 0.240 | 0.181 | 0.281 | 0.140 | 0.237 |
| | 192 | 0.155 | 0.251 | **0.147** | **0.240** | 0.157 | 0.253 | 0.193 | 0.293 | 0.153 | 0.249 |
| | 336 | 0.172 | 0.268 | **0.163** | **0.257** | 0.174 | 0.270 | 0.205 | 0.312 | 0.169 | 0.267 |
| | 720 | 0.211 | 0.300 | **0.198** | **0.290** | 0.212 | 0.301 | 0.222 | 0.320 | 0.203 | 0.301 |
| ILI | 24 | 1.987 | 0.906 | **1.985** | **0.905** | 2.031 | 0.934 | 2.500 | 1.055 | 2.215 | 1.081 |
| | 36 | 1.928 | 0.915 | 1.897 | **0.900** | **1.889** | 0.902 | 2.222 | 1.007 | 1.963 | 0.963 |
| | 48 | 1.904 | 0.919 | **1.868** | **0.910** | 1.902 | 0.922 | 2.304 | 1.043 | 2.130 | 1.024 |
| | 60 | 2.066 | 0.976 | **1.928** | **0.933** | 2.012 | 0.958 | 2.354 | 1.046 | 2.368 | 1.096 |

∗ We replace the input length L=512 in TimesNet and MICN for a fair comparison.

**CNN Kernel**    The convolution kernel size determines the receptive field for extracting periodicity from the sequence. In this section, we consider the model performance with different kernel size to {3, 5, 7}, respectively, and the experimental results are shown in Appendix Table 10. It can be found that a larger convolution kernel causes lower prediction accuracy. The results can be attributed that after the proposed period decomposition, the period of the reshaped sequence is smaller and a larger kernel excessively extracts the temporal correlation from other periods to degrade the model performance, especially in the large datasets, traffic and ECL.

Table 10: Prediction error (MSE & MAE) with different kernel size.

| Methods | | WinNet | | | | | | TimesNet* | | DLinear | |
|---|---|---|---|---|---|---|---|---|---|---|---|
| | | 3x3 | | 5x5 | | 7x7 | | | | | |
| Metric | | MSE | MAE | MSE | MAE | MSE | MAE | MSE | MAE | MSE | MAE |
| ETTm1 | 96 | **0.283** | **0.335** | 0.288 | 0.339 | 0.286 | 0.338 | 0.377 | 0.397 | 0.290 | 0.342 |
| | 192 | **0.324** | **0.360** | 0.335 | 0.368 | 0.329 | 0.364 | 0.389 | 0.401 | 0.332 | 0.369 |
| | 336 | **0.357** | **0.379** | 0.370 | 0.385 | 0.366 | 0.380 | 0.393 | 0.414 | 0.366 | 0.392 |
| | 720 | **0.416** | **0.411** | 0.423 | 0.414 | 0.422 | 0.413 | 0.470 | 0.458 | 0.420 | 0.424 |
| ETTm2 | 96 | **0.160** | **0.251** | 0.163 | 0.252 | 0.162 | 0.252 | 0.201 | 0.280 | 0.167 | 0.260 |
| | 192 | **0.212** | **0.287** | 0.216 | 0.289 | 0.217 | 0.290 | 0.242 | 0.313 | 0.224 | 0.303 |
| | 336 | **0.261** | 0.322 | 0.267 | 0.321 | 0.269 | 0.324 | 0.310 | 0.356 | 0.281 | 0.342 |
| | 720 | **0.359** | **0.381** | 0.361 | 0.382 | 0.375 | 0.387 | 0.381 | 0.396 | 0.397 | 0.421 |
| ETTh1 | 96 | **0.362** | **0.390** | 0.368 | 0.392 | 0.366 | 0.392 | 0.460 | 0.464 | 0.375 | 0.399 |
| | 192 | **0.394** | **0.410** | 0.410 | 0.420 | 0.425 | 0.434 | 0.458 | 0.462 | 0.412 | 0.420 |
| | 336 | **0.419** | **0.426** | 0.436 | 0.434 | 0.442 | 0.442 | 0.523 | 0.501 | 0.439 | 0.443 |
| | 720 | 0.436 | **0.453** | **0.435** | **0.453** | 0.436 | **0.453** | 0.502 | 0.497 | 0.472 | 0.490 |
| ETTh2 | 96 | **0.267** | **0.332** | 0.274 | 0.337 | 0.270 | 0.334 | 0.338 | 0.397 | 0.289 | 0.353 |
| | 192 | **0.322** | **0.372** | 0.337 | 0.378 | 0.360 | 0.399 | 0.422 | 0.446 | 0.383 | 0.418 |
| | 336 | **0.351** | **0.401** | 0.376 | 0.409 | 0.364 | 0.409 | 0.431 | 0.460 | 0.448 | 0.465 |
| | 720 | **0.389** | **0.436** | 0.502 | 0.532 | 0.418 | 0.458 | 0.467 | 0.480 | 0.605 | 0.551 |
| weather | 96 | 0.143 | **0.198** | **0.141** | 0.239 | 0.150 | 0.203 | 0.163 | 0.223 | 0.176 | 0.237 |
| | 192 | **0.188** | **0.240** | 0.193 | 0.245 | 0.193 | 0.244 | 0.218 | 0.266 | 0.192 | 0.246 |
| | 336 | **0.235** | **0.280** | 0.243 | 0.285 | 0.244 | 0.287 | 0.280 | 0.306 | 0.240 | 0.287 |
| | 720 | **0.310** | **0.336** | 0.323 | 0.346 | 0.318 | 0.340 | 0.349 | 0.356 | 0.316 | 0.352 |
| ECL | 96 | **0.130** | **0.226** | 0.141 | 0.239 | 0.140 | 0.238 | 0.181 | 0.281 | 0.140 | 0.237 |
| | 192 | **0.147** | **0.240** | 0.161 | 0.258 | 0.161 | 0.258 | 0.193 | 0.293 | 0.153 | 0.249 |
| | 336 | **0.163** | **0.257** | 0.178 | 0.274 | 0.178 | 0.274 | 0.205 | 0.312 | 0.169 | 0.267 |
| | 720 | **0.198** | **0.290** | 0.212 | 0.301 | 0.216 | 0.306 | 0.222 | 0.320 | 0.203 | 0.301 |
| traffic | 96 | **0.394** | **0.274** | 0.414 | 0.287 | 0.415 | 0.289 | 0.603 | 0.328 | 0.410 | 0.282 |
| | 192 | **0.407** | **0.279** | 0.426 | 0.294 | 0.428 | 0.295 | 0.610 | 0.329 | 0.423 | 0.287 |
| | 336 | **0.416** | **0.283** | 0.436 | 0.295 | 0.439 | 0.297 | 0.619 | 0.330 | 0.436 | 0.296 |
| | 720 | **0.453** | **0.305** | 0.464 | 0.310 | 0.467 | 0.313 | 0.632 | 0.352 | 0.466 | 0.315 |
| ILI | 24 | **1.985** | **0.905** | 2.034 | 0.923 | 1.966 | 0.900 | 2.500 | 1.055 | 2.215 | 1.081 |
| | 36 | **1.897** | **0.900** | 1.906 | 0.909 | 1.900 | 0.902 | 2.222 | 1.007 | 1.963 | 0.963 |
| | 48 | **1.868** | **0.910** | 1.925 | 0.930 | 1.906 | 0.922 | 2.304 | 1.043 | 2.130 | 1.024 |
| | 60 | **1.928** | **0.933** | 1.937 | 0.934 | 1.945 | 0.937 | 2.354 | 1.046 | 2.368 | 1.096 |

∗ We replace the input length L=512 in TimesNet for a fair comparison.

**Fusion mode**    In the Series Decoder block, we need to fuse the periodic features extracted by DCB for both intra-period and inter-period. Since they are both square shapes and the inter-period is obtained from the transpose of the intra-period, there are two ways to fuse them: one is to transpose the inter-period back and sum them up, and the other is to sum them up directly. Therefore, we explore the impact of these two fusion modes on the model performance. As shown in Appendix Table 11, we can assume that superior results can be achieved by using direct fusing. Even with the NF approach, WinNet almost outperforms CNN-based TimesNet and MLP-based DLinear.

**Intra-Inter period**    In the WinNet, we get two inputs based on the I2PE block, i.e., intra-period and inter-period. The intra-period represents the local periodicity by rows, while inter-period represents the local periodicity by columns, and a parameter-shared convolutional kernel is utilized to

Table 11: Ablation study the fusion mode of inter-period and intra-period on all datasets in WinNet. NF means the normal fusion, while DF means no transposition, direct fusion.

| Methods | | WinNet | | | | TimesNet | | DLinear | |
|---|---|---|---|---|---|---|---|---|---|
| | | DF | | NF | | | | | |
| Metric | | MSE | MAE | MSE | MAE | MSE | MAE | MSE | MAE |
| ETTm1 | 96 | **0.283** | **0.335** | 0.286 | 0.337 | 0.377 | 0.397 | 0.290 | 0.342 |
| | 192 | **0.324** | **0.360** | 0.326 | 0.360 | 0.389 | 0.401 | 0.332 | 0.369 |
| | 336 | **0.357** | **0.379** | 0.366 | 0.381 | 0.393 | 0.414 | 0.366 | 0.392 |
| | 720 | **0.416** | **0.411** | 0.421 | 0.414 | 0.470 | 0.458 | 0.420 | 0.424 |
| ETTm2 | 96 | **0.160** | **0.251** | 0.162 | 0.252 | 0.201 | 0.280 | 0.167 | 0.260 |
| | 192 | **0.212** | **0.287** | 0.217 | 0.290 | 0.242 | 0.313 | 0.224 | 0.303 |
| | 336 | **0.261** | **0.322** | 0.273 | 0.326 | 0.310 | 0.356 | 0.281 | 0.342 |
| | 720 | **0.359** | **0.381** | 0.363 | 0.384 | 0.381 | 0.396 | 0.397 | 0.421 |
| ETTh1 | 96 | **0.362** | **0.390** | 0.369 | 0.393 | 0.460 | 0.464 | 0.375 | 0.399 |
| | 192 | **0.394** | **0.410** | 0.409 | 0.419 | 0.458 | 0.462 | 0.412 | 0.420 |
| | 336 | **0.419** | **0.426** | 0.432 | 0.432 | 0.523 | 0.501 | 0.439 | 0.443 |
| | 720 | **0.436** | **0.453** | 0.438 | 0.456 | 0.502 | 0.497 | 0.472 | 0.490 |
| ETTh2 | 96 | **0.267** | **0.332** | 0.270 | 0.335 | 0.338 | 0.397 | 0.289 | 0.353 |
| | 192 | **0.322** | **0.372** | 0.330 | 0.376 | 0.422 | 0.446 | 0.383 | 0.418 |
| | 336 | **0.351** | **0.401** | 0.385 | 0.413 | 0.431 | 0.460 | 0.448 | 0.465 |
| | 720 | **0.389** | **0.436** | 0.424 | 0.464 | 0.467 | 0.480 | 0.605 | 0.551 |
| Weather | 96 | **0.143** | **0.198** | 0.154 | 0.206 | 0.163 | 0.223 | 0.176 | 0.237 |
| | 192 | **0.188** | **0.240** | 0.193 | 0.246 | 0.218 | 0.266 | 0.192 | 0.246 |
| | 336 | **0.235** | **0.280** | 0.244 | 0.285 | 0.280 | 0.306 | 0.240 | 0.287 |
| | 720 | **0.310** | **0.336** | 0.321 | 0.342 | 0.349 | 0.356 | 0.316 | 0.352 |
| ECL | 96 | **0.130** | **0.226** | 0.141 | 0.239 | 0.181 | 0.281 | 0.140 | 0.237 |
| | 192 | **0.147** | **0.240** | 0.156 | 0.252 | 0.193 | 0.293 | 0.153 | 0.249 |
| | 336 | **0.163** | **0.257** | 0.173 | 0.269 | 0.205 | 0.312 | 0.169 | 0.267 |
| | 720 | **0.198** | **0.290** | 0.211 | 0.300 | 0.222 | 0.320 | 0.203 | 0.301 |
| traffic | 96 | **0.394** | **0.274** | 0.414 | 0.288 | 0.603 | 0.328 | 0.410 | 0.282 |
| | 192 | **0.407** | **0.279** | 0.427 | 0.294 | 0.610 | 0.329 | 0.423 | 0.287 |
| | 336 | **0.416** | **0.283** | 0.435 | 0.296 | 0.619 | 0.330 | 0.436 | 0.296 |
| | 720 | **0.453** | **0.305** | 0.465 | 0.312 | 0.632 | 0.352 | 0.466 | 0.315 |
| ILI | 24 | **1.985** | **0.905** | 2.003 | 0.923 | 2.500 | 1.055 | 2.215 | 1.081 |
| | 36 | **1.897** | **0.900** | 1.906 | 0.909 | 2.222 | 1.007 | 1.963 | 0.963 |
| | 48 | **1.868** | **0.910** | 1.925 | 0.930 | 2.304 | 1.043 | 2.130 | 1.024 |
| | 60 | **1.928** | **0.933** | 1.937 | 0.934 | 2.354 | 1.046 | 2.368 | 1.096 |

∗ We replace the input length L=512 in TimesNet for a fair comparison.

extract the period variations of each of these two periods. Therefore, we design an ablation study to validate their respective effects on the model prediction ability. From Appendix Table 12, it can be seen that a single period may achieve comparable results against the double period on some datasets, such as weather. However, the single period is less effective on multi-channel large datasets like ECL and traffic.

Table 12: Prediction error (MSE & MAE) with the ablation studies of the inter-intra period. Full means both intra-period and inter-period are used to extract periodic features. Intra means only intra-period inputs are used and inter means only inter-period are used.

| Methods | | WinNet | | | | | |
|---------|---|---|---|---|---|---|---|
| | | Inter+Intra | | Intra | | Inter | |
| Metric | | MSE | MAE | MSE | MAE | MSE | MAE |
| ETTm1 | 96 | **0.283** | **0.335** | 0.287 | 0.338 | 0.328 | 0.365 |
| | 192 | **0.324** | **0.360** | 0.329 | 0.361 | 0.331 | 0.362 |
| | 336 | **0.357** | **0.379** | 0.362 | 0.380 | 0.366 | 0.383 |
| | 720 | **0.416** | **0.411** | 0.417 | 0.410 | 0.418 | **0.411** |
| ETTm2 | 24 | **0.160** | 0.251 | 0.161 | **0.250** | 0.164 | 0.252 |
| | 36 | **0.212** | **0.287** | 0.217 | 0.289 | 0.219 | 0.291 |
| | 48 | **0.261** | **0.322** | 0.269 | 0.323 | 0.275 | 0.328 |
| | 60 | **0.359** | **0.381** | 0.360 | 0.382 | 0.360 | 0.382 |
| ETTh1 | 96 | **0.362** | **0.390** | 0.378 | 0.402 | 0.368 | 0.396 |
| | 192 | **0.394** | **0.410** | 0.412 | 0.426 | 0.404 | 0.420 |
| | 336 | **0.419** | **0.426** | 0.436 | 0.442 | 0.426 | 0.434 |
| | 720 | **0.436** | **0.453** | 0.451 | 0.466 | 0.445 | 0.461 |
| ETTh2 | 96 | **0.267** | **0.332** | 0.267 | 0.333 | 0.281 | 0.340 |
| | 192 | **0.322** | **0.372** | 0.327 | 0.376 | 0.347 | 0.388 |
| | 336 | **0.351** | **0.401** | 0.360 | 0.404 | 0.367 | 0.410 |
| | 720 | **0.389** | **0.436** | 0.402 | 0.440 | 0.397 | 0.439 |
| ILI | 24 | **1.985** | **0.905** | 2.188 | 1.008 | 2.177 | 1.006 |
| | 36 | **1.897** | **0.900** | 1.946 | 0.917 | 1.935 | 0.915 |
| | 48 | **1.868** | **0.910** | 1.926 | 0.932 | 1.916 | 0.930 |
| | 60 | **1.928** | **0.933** | 1.951 | 0.943 | 1.947 | 0.942 |
| ECL | 96 | **0.130** | **0.226** | 0.137 | 0.235 | 0.139 | 0.237 |
| | 192 | **0.147** | **0.240** | 0.154 | 0.251 | 0.157 | 0.255 |
| | 336 | **0.163** | **0.257** | 0.171 | 0.268 | 0.174 | 0.273 |
| | 720 | **0.198** | **0.290** | 0.210 | 0.300 | 0.211 | 0.301 |
| traffic | 96 | **0.394** | **0.274** | 0.413 | 0.301 | 0.412 | 0.302 |
| | 192 | **0.407** | **0.279** | 0.440 | 0.322 | 0.438 | 0.320 |
| | 336 | **0.416** | **0.283** | 0.436 | 0.311 | 0.434 | 0.309 |
| | 720 | **0.453** | **0.305** | 0.487 | 0.345 | 0.487 | 0.347 |
| weather | 96 | **0.143** | 0.198 | 0.144 | **0.196** | 0.152 | 0.205 |
| | 192 | **0.188** | **0.240** | 0.188 | 0.242 | 0.193 | 0.242 |
| | 336 | **0.235** | **0.280** | 0.238 | 0.283 | 0.244 | 0.286 |
| | 720 | **0.310** | **0.336** | 0.310 | 0.339 | 0.312 | 0.338 |
| Exchange | 96 | **0.085** | **0.201** | 0.085 | 0.202 | 0.085 | 0.202 |
| | 192 | 0.180 | **0.298** | 0.180 | 0.300 | **0.179** | **0.298** |
| | 336 | **0.307** | **0.398** | 0.349 | 0.421 | 0.344 | 0.422 |
| | 720 | **1.032** | **0.761** | 1.142 | 0.782 | 1.130 | 0.785 |

**Model architecture**  We propose three blocks: I2PE, TDPD and DCB. Here, we validate the performance of these 3 modules on the ETT dataset and compare with TimesNet and DLinear.

- original: We process the sequence into the trend-cyclical and seasonal terms by common trend decomposition from DLinear and then extract features by common convolution network, respectively.

- TDPD: We optimise the trend decomposition on the sequence by the TDPD module and also extract features by common convolution network, respectively.

- TDPD+DCB: We utilize the decomposition strategy by the TDPD module and together extract periodic features of the period-trend and oscillation terms by DCB block.

- Final: Unlike the previous ones that used a single input, we design both intra-periodic and inter-periodic inputs into the TDPD and DCB modules, respectively. Combine the extracted features by simply adding them.

Table 13: Ablation study on all ETT datasets of the proposed modules including the I2PE, TDPD and DCB in WinNet. Four cases are included: (a) all the three modules are included in model (Final: I2PE+TDPD+DCB); (b) only the TDPD; (c) TDPD+DCB; (d) the original version with the common CNN and one-dimensional trend decomposition.

| Methods | | WinNet | | | | | | | | TimesNet | | DLinear | |
|---|---|---|---|---|---|---|---|---|---|---|---|---|---|
| | | Final | | TDPD+DCB | | TDPD | | original | | | | | |
| Metric | | MSE | MAE | MSE | MAE | MSE | MAE | MSE | MAE | MSE | MAE | MSE | MAE |
| ETTm1 | 96 | **0.284** | **0.338** | 0.288 | 0.341 | 0.289 | 0.341 | 0.292 | 0.352 | 0.377 | 0.397 | 0.290 | 0.342 |
| | 192 | **0.326** | **0.362** | 0.328 | 0.363 | 0.330 | 0.365 | 0.327 | 0.374 | 0.389 | 0.401 | 0.332 | 0.369 |
| | 336 | **0.359** | **0.380** | 0.362 | 0.385 | 0.363 | 0.384 | 0.357 | 0.391 | 0.393 | 0.414 | 0.366 | 0.392 |
| | 720 | 0.420 | **0.411** | **0.409** | 0.413 | 0.428 | 0.421 | **0.409** | 0.423 | 0.470 | 0.458 | 0.420 | 0.424 |
| ETTm2 | 96 | 0.160 | 0.251 | **0.159** | **0.247** | 0.161 | 0.249 | 0.359 | 0.390 | 0.201 | 0.280 | 0.167 | 0.260 |
| | 192 | **0.212** | **0.287** | 0.217 | 0.288 | 0.217 | 0.290 | 0.611 | 0.511 | 0.242 | 0.313 | 0.224 | 0.303 |
| | 336 | **0.261** | **0.322** | 0.270 | 0.326 | 0.268 | 0.324 | 0.855 | 0.613 | 0.310 | 0.356 | 0.281 | 0.342 |
| | 720 | 0.359 | 0.381 | **0.358** | **0.380** | 0.359 | 0.383 | 1.002 | 0.674 | 0.381 | 0.396 | 0.397 | 0.421 |
| ETTh1 | 96 | **0.362** | **0.390** | 0.363 | 0.393 | 0.377 | 0.404 | 0.379 | 0.415 | 0.460 | 0.464 | 0.375 | 0.399 |
| | 192 | **0.394** | **0.410** | 0.397 | 0.415 | 0.402 | 0.418 | 0.411 | 0.438 | 0.458 | 0.462 | 0.412 | 0.420 |
| | 336 | **0.420** | **0.427** | 0.422 | 0.431 | 0.448 | 0.450 | 0.440 | 0.461 | 0.523 | 0.501 | 0.439 | 0.443 |
| | 720 | **0.436** | **0.455** | 0.437 | 0.456 | 0.449 | 0.461 | 0.506 | 0.525 | 0.502 | 0.497 | 0.472 | 0.490 |
| ETTh2 | 96 | 0.267 | **0.332** | **0.264** | **0.332** | 0.269 | 0.333 | 0.842 | 0.622 | 0.338 | 0.397 | 0.289 | 0.353 |
| | 192 | 0.322 | **0.372** | **0.319** | **0.372** | 0.330 | 0.375 | 0.971 | 0.682 | 0.422 | 0.446 | 0.383 | 0.418 |
| | 336 | 0.351 | **0.401** | **0.349** | **0.401** | 0.382 | 0.416 | 1.098 | 0.728 | 0.431 | 0.460 | 0.448 | 0.465 |
| | 720 | **0.389** | **0.436** | 0.391 | 0.437 | 0.398 | 0.443 | 1.428 | 0.854 | 0.467 | 0.480 | 0.605 | 0.551 |

∗ We replace the input length L=512 in TimesNet for a fair comparison.

Table 14: Efficiency of our model on the ECL dataset vs other methods in multivariate prediction. We set the input length to 720 and the prediction length to 720. See relevant computational efficiency with thop, torchsummary and torch.cuda.memory_allocated functions. Times-T denotes the time of an iter training, and Times-I denotes the inference time. In addition, f. means former.

| Method | WinNet | PatchTST | TimesNet | MICN | Crossf. | DLinear | FEDf. | Autof. | Inf. | Transf. |
|---|---|---|---|---|---|---|---|---|---|---|
| FLOPs | 273M | 51.1G | 1620.1G | 5.95G | 146.4G | 333M | 2.35G | 2.35G | 1.84G | 2.17G |
| Parameter | 836.8K | 18.18M | 226.3M | 19.07M | 11.1M | 1.04M | 4.76M | 3.19M | 3.35M | 2.95M |
| Times-T | 0.42s | 0.17s | 0.70s | 0.04s | 0.54s | 0.01s | 0.56s | 0.26s | 0.14s | 0.03s |
| Memory | 12MiB | 85MiB | 878MiB | 88MiB | 96MiB | 14MiB | 39MiB | 32MiB | 33MiB | 31MiB |
| Times-I | 61.3ms | 33.3ms | 115.1ms | 20.2ms | 66.8ms | 19.9ms | 89.4ms | 38.9ms | 32.3ms | 36.5ms |

Table 15: Efficiency of our model on the ETTm1 dataset vs. other methods in multivariate prediction. We set the input length to 720 and the prediction length to 720. See relevant computational efficiency with thop, torchsummary and torch.cuda.memory_allocated functions. Times-T denotes the time of an iter training, and Times-I denotes the inference time. In addition, f. means former.

| Method | WinNet | PatchTST | TimesNet | MICN | Crossf. | DLinear | FEDf. | Autof. | Inf. | Transf. |
|---|---|---|---|---|---|---|---|---|---|---|
| FLOPs | 5.96M | 309M | 406.2G | 5.33G | 3.46G | 7.26M | 1.75G | 1.75G | 1.42G | 1.75G |
| Params | 830.9K | 8.69M | 57.28M | 18.75M | 11.09M | 1.04M | 3.96M | 2.39M | 2.78M | 2.39M |
| Times-T | 28ms | 26ms | 500ms | 25ms | 62ms | 15ms | 450ms | 270ms | 150ms | 45ms |
| Memory | 11MiB | 44MiB | 239MiB | 85MiB | 57MiB | 12MiB | 24MiB | 27MiB | 29MiB | 27MiB |
| Times-I | 9.8ms | 10.6ms | 65.2ms | 11.6ms | 22.0ms | 9.1ms | 61.4ms | 26.5ms | 17.4ms | 14.2ms |

### A.3 PREDICTION VISUALIZATION

As shown in Appendix Figure 10, Figure 11, Figure 12, Figure 13, Figure 14, Figure 15, Figure 16, we visualize the long-term univariate prediction results of WinNet from the test set of all eight datasets. Here, we predict {24, 36, 48, 60} steps on ILI dataset and {96, 192, 336, 720} steps on other datasets. It can be seen that the proposed model can achieve the best results. We also supplement the performance with different look-back windows on ETTs, as shown in Figure 9.

Figure 9: Prediction error (MSE) with different look-back windows on three ETT datasets: ETTm1, ETTm2, ETTh1. The look-back windows are selected to be L = {24, 48, 96, 192, 336, 512, 720}, and the prediction lengths are T = {96, 720}.

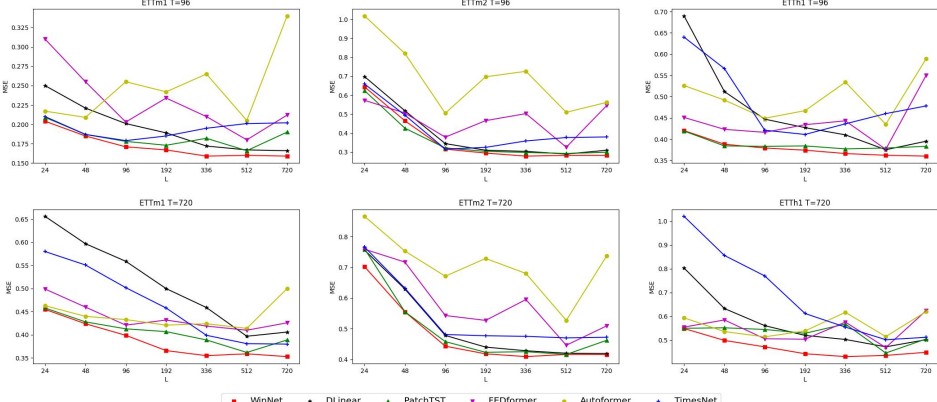

Figure 10: Visualization of prediction on traffic with the look-back window L=336.

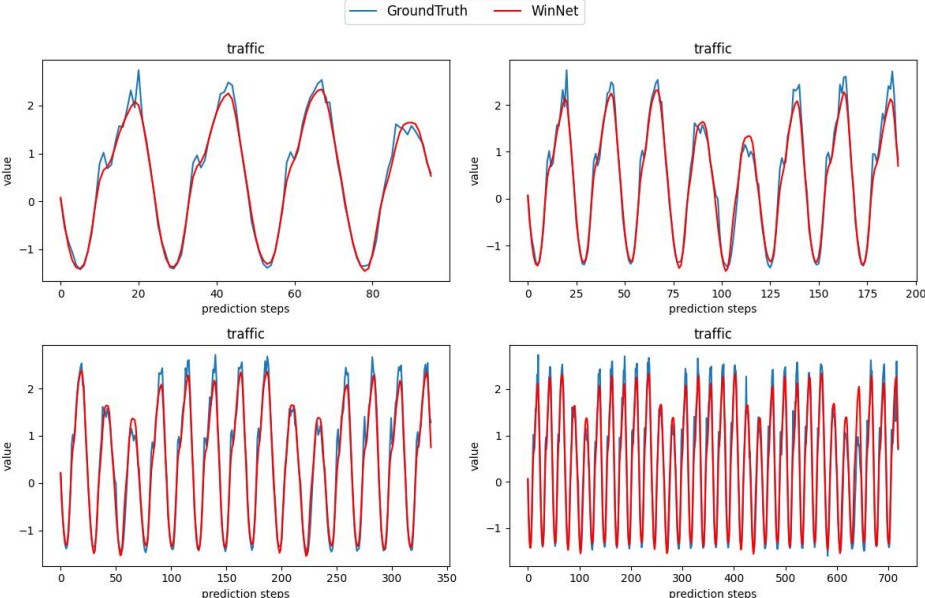

Figure 11: Visualization of prediction on ETTm2 with the look-back window L=336.

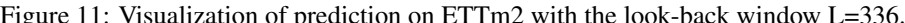

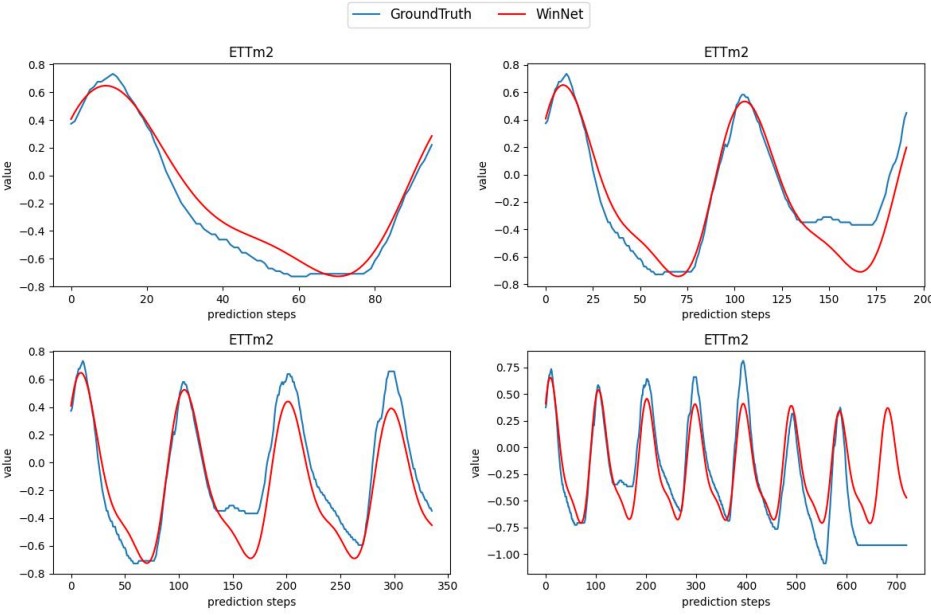

Figure 12: Visualization of prediction on ETTh1 with the look-back window L=336.

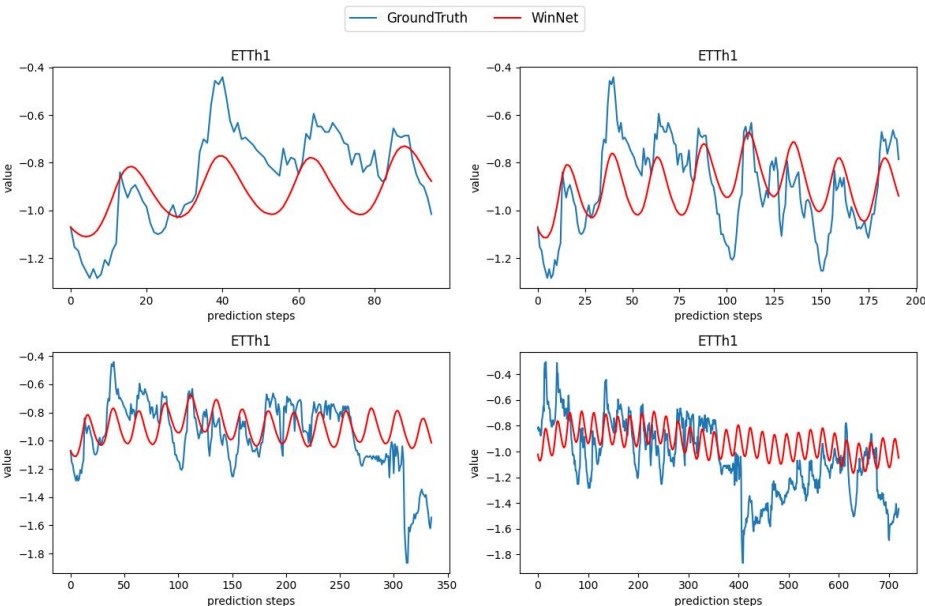

Figure 13: Visualization of prediction on ECL with the look-back window L=336.

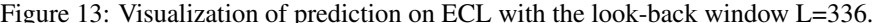

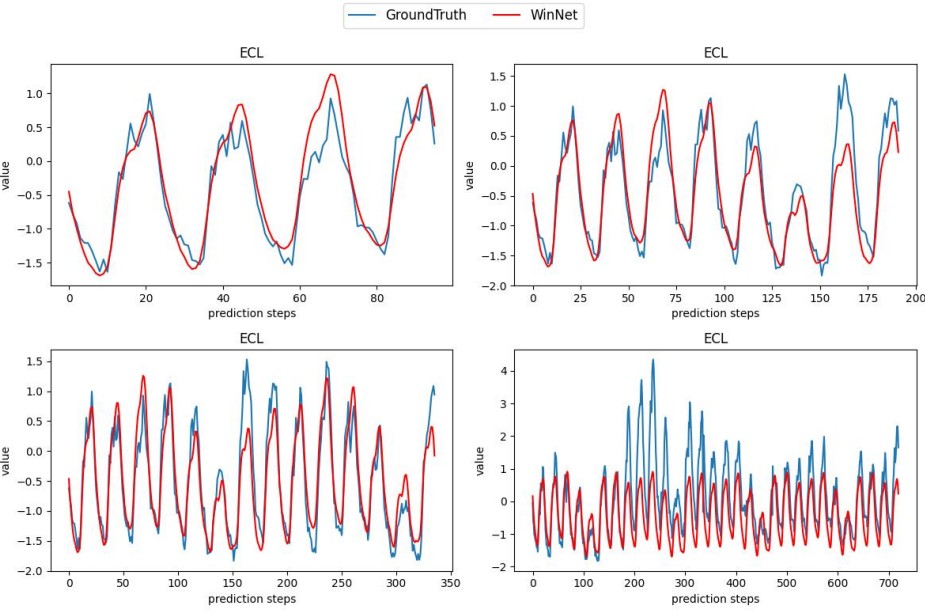

Figure 14: Visualization of prediction on ETTh2 with the look-back window L=336.

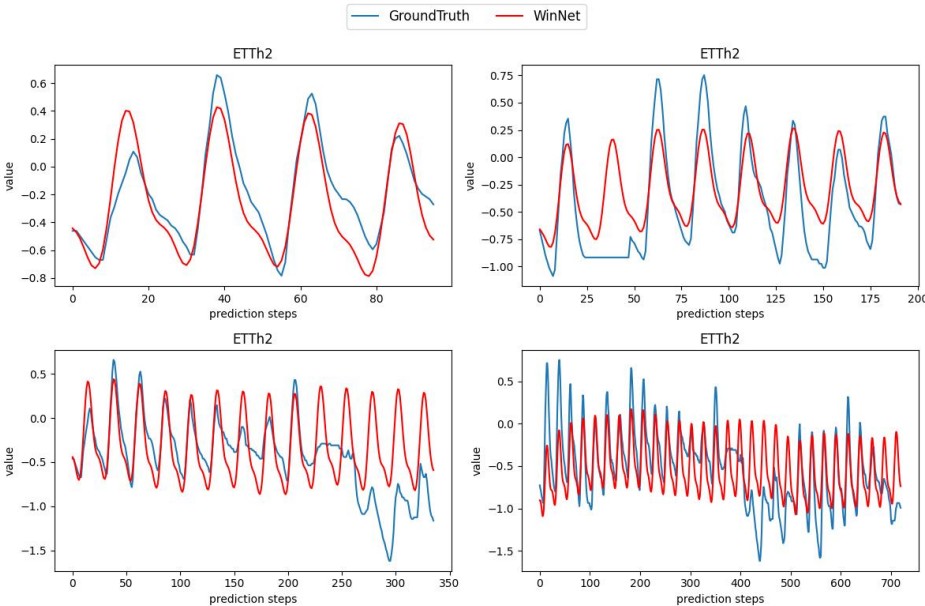

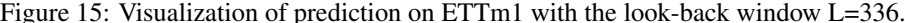

Figure 15: Visualization of prediction on ETTm1 with the look-back window L=336.

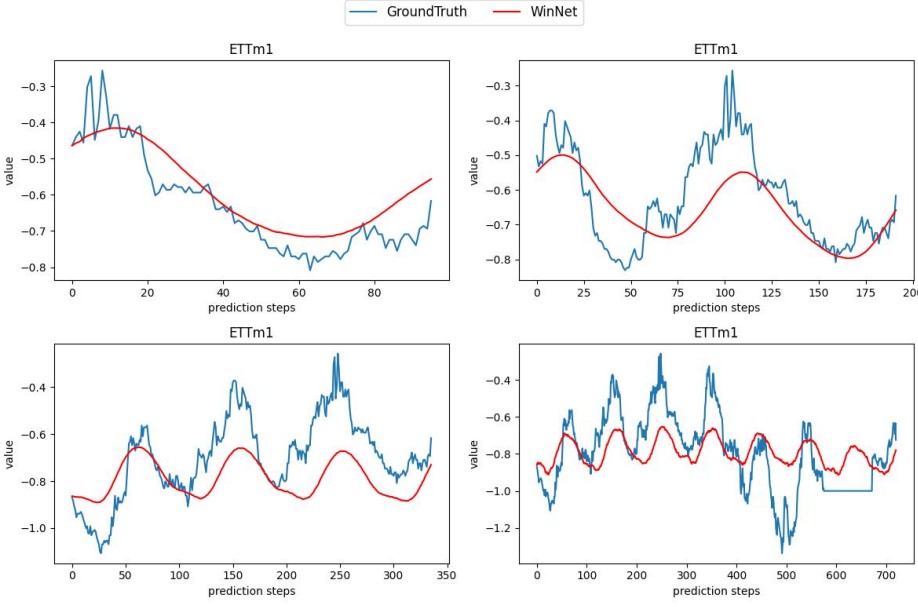

Figure 16: Visualization of prediction on ILI with the look-back window L=104.

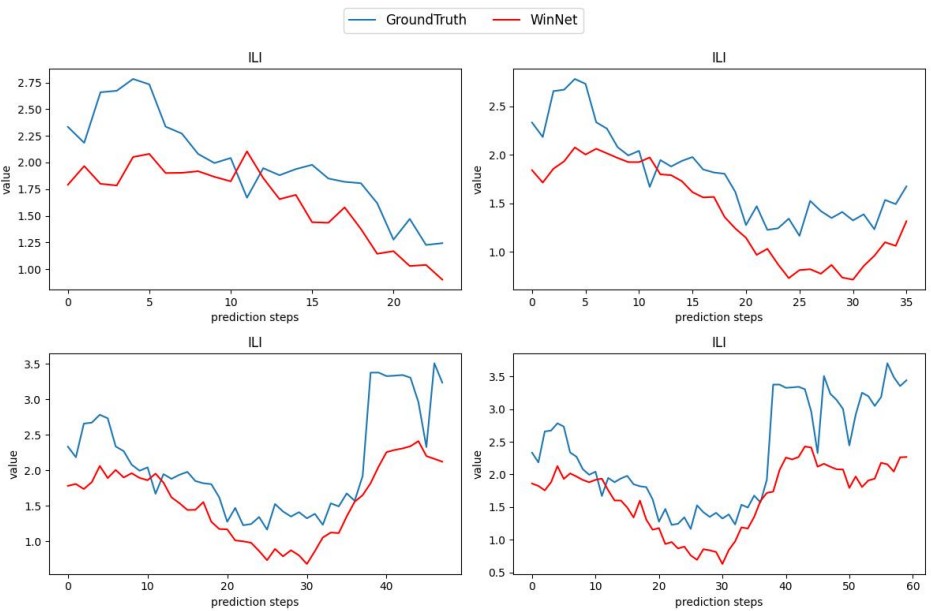

## A.4 FULL RESULTS

Due to the space limitation of the main text, we place the full results of all experiments in the following: multivariate long-input length forecasting in Appendix Table 16, multivariate short-input length forecasting in Appendix Table 17, univariate long-input length forecasting in Appendix Table 18.

Table 16: Full results for multivariate long-input length prediction. We compare extensive competitive models under different prediction lengths. The input sequence length is set to 104 for the ILI dataset and 512 for the others. The 1st count indicates the numbers of best performance.

| Methods | | WinNet (Ours) | | RLinear (2023a) | | RMLP (2023a) | | PatchTST (2023) | | TimesNet (2023) | | MICN (2023) | | Crossformer (2023) | | DLinear (2023) | | FEDformer (2022) | | Autoformer (2021) | |
|---|---|---|---|---|---|---|---|---|---|---|---|---|---|---|---|---|---|---|---|---|---|
| Metric | | MSE | MAE | MSE | MAE | MSE | MAE | MSE | MAE | MSE | MAE | MSE | MAE | MSE | MAE | MSE | MAE | MSE | MAE | MSE | MAE |
| ETTm1 | 96 | **0.283** | **0.335** | 0.313 | 0.358 | 0.312 | 0.362 | 0.293 | 0.346 | 0.377 | 0.397 | 0.305 | 0.354 | 0.302 | 0.359 | 0.290 | 0.342 | 0.326 | 0.390 | 0.510 | 0.492 |
| | 192 | **0.324** | **0.360** | 0.339 | 0.371 | 0.344 | 0.386 | 0.333 | 0.370 | 0.389 | 0.401 | 0.362 | 0.399 | 0.341 | 0.387 | 0.332 | 0.369 | 0.365 | 0.415 | 0.514 | 0.495 |
| | 336 | **0.357** | **0.379** | 0.441 | 0.460 | 0.381 | 0.409 | 0.369 | 0.392 | 0.393 | 0.414 | 0.382 | 0.405 | 0.419 | 0.432 | 0.366 | 0.392 | 0.392 | 0.425 | 0.510 | 0.492 |
| | 720 | **0.416** | **0.411** | 0.420 | 0.418 | 0.432 | 0.433 | **0.416** | 0.420 | 0.470 | 0.458 | 0.441 | 0.441 | 0.637 | 0.577 | 0.420 | 0.424 | 0.446 | 0.458 | 0.527 | 0.493 |
| | Avg | **0.345** | **0.371** | 0.378 | 0.402 | 0.367 | 0.398 | 0.352 | 0.382 | 0.407 | 0.417 | 0.372 | 0.399 | 0.424 | 0.438 | 0.352 | 0.381 | 0.382 | 0.422 | 0.515 | 0.493 |
| ETTm2 | 96 | **0.160** | **0.251** | 0.169 | 0.266 | 0.184 | 0.269 | 0.166 | 0.256 | 0.201 | 0.280 | 0.189 | 0.287 | 0.305 | 0.361 | 0.167 | 0.260 | 0.180 | 0.271 | 0.205 | 0.293 |
| | 192 | **0.212** | **0.287** | 0.258 | 0.333 | 0.248 | 0.325 | 0.223 | 0.296 | 0.242 | 0.313 | 0.239 | 0.323 | 0.355 | 0.391 | 0.224 | 0.303 | 0.252 | 0.318 | 0.278 | 0.336 |
| | 336 | **0.261** | **0.322** | 0.301 | 0.358 | 0.327 | 0.382 | 0.274 | 0.329 | 0.310 | 0.356 | 0.348 | 0.385 | 0.420 | 0.431 | 0.281 | 0.342 | 0.324 | 0.364 | 0.343 | 0.379 |
| | 720 | **0.359** | **0.381** | 0.397 | 0.426 | 0.406 | 0.425 | 0.362 | 0.385 | 0.381 | 0.396 | 0.421 | 0.434 | 0.592 | 0.548 | 0.397 | 0.421 | 0.410 | 0.420 | 0.414 | 0.419 |
| | Avg | **0.248** | **0.310** | 0.281 | 0.346 | 0.291 | 0.350 | 0.256 | 0.316 | 0.283 | 0.336 | 0.299 | 0.357 | 0.418 | 0.432 | 0.267 | 0.331 | 0.291 | 0.343 | 0.310 | 0.356 |
| ETTh1 | 96 | **0.362** | **0.390** | 0.371 | 0.397 | 0.401 | 0.430 | 0.379 | 0.401 | 0.460 | 0.464 | 0.404 | 0.429 | 0.394 | 0.418 | 0.375 | 0.399 | 0.376 | 0.415 | 0.435 | 0.446 |
| | 192 | **0.394** | **0.410** | 0.422 | 0.439 | 0.427 | 0.441 | 0.413 | 0.429 | 0.458 | 0.462 | 0.511 | 0.506 | 0.423 | 0.436 | 0.412 | 0.420 | 0.423 | 0.446 | 0.456 | 0.457 |
| | 336 | **0.419** | **0.426** | 0.490 | 0.488 | 0.469 | 0.471 | 0.435 | 0.436 | 0.523 | 0.501 | 0.482 | 0.489 | 0.438 | 0.451 | 0.439 | 0.443 | 0.444 | 0.462 | 0.486 | 0.487 |
| | 720 | **0.436** | **0.453** | 0.484 | 0.500 | 0.545 | 0.528 | 0.446 | 0.464 | 0.502 | 0.497 | 0.697 | 0.631 | 0.508 | 0.514 | 0.472 | 0.490 | 0.469 | 0.492 | 0.515 | 0.517 |
| | Avg | **0.402** | **0.419** | 0.442 | 0.456 | 0.461 | 0.468 | 0.418 | 0.432 | 0.485 | 0.481 | 0.523 | 0.513 | 0.440 | 0.454 | 0.424 | 0.438 | 0.428 | 0.453 | 0.473 | 0.476 |
| ETTh2 | 96 | **0.267** | **0.332** | 0.288 | 0.355 | 0.304 | 0.369 | 0.274 | 0.337 | 0.338 | 0.397 | 0.290 | 0.356 | 0.395 | 0.417 | 0.289 | 0.353 | 0.332 | 0.374 | 0.332 | 0.368 |
| | 192 | **0.322** | **0.372** | 0.377 | 0.415 | 0.381 | 0.418 | 0.338 | 0.376 | 0.422 | 0.446 | 0.415 | 0.441 | 0.427 | 0.438 | 0.383 | 0.418 | 0.407 | 0.446 | 0.426 | 0.434 |
| | 336 | **0.351** | **0.401** | 0.426 | 0.453 | 0.401 | 0.440 | 0.363 | 0.397 | 0.431 | 0.460 | 0.627 | 0.573 | 0.449 | 0.459 | 0.448 | 0.465 | 0.400 | 0.447 | 0.477 | 0.479 |
| | 720 | **0.389** | 0.436 | 0.783 | 0.627 | 0.615 | 0.564 | 0.393 | **0.430** | 0.467 | 0.480 | 1.340 | 0.858 | 0.501 | 0.509 | 0.605 | 0.551 | 0.412 | 0.469 | 0.453 | 0.490 |
| | Avg | **0.332** | **0.385** | 0.469 | 0.463 | 0.425 | 0.448 | 0.342 | **0.385** | 0.414 | 0.445 | 0.668 | 0.557 | 0.443 | 0.455 | 0.431 | 0.446 | 0.387 | 0.434 | 0.422 | 0.442 |
| ILI | 24 | 1.985 | 0.905 | 2.404 | 1.142 | 2.180 | 1.046 | **1.522** | **0.814** | 2.500 | 1.055 | 2.559 | 1.099 | 3.383 | 1.249 | 2.215 | 1.081 | 2.624 | 1.095 | 2.906 | 1.182 |
| | 36 | 1.897 | 0.900 | 2.378 | 1.097 | 2.272 | 1.054 | **1.430** | **0.834** | 2.222 | 1.007 | 2.483 | 1.023 | 3.151 | 1.157 | 1.963 | 0.963 | 2.516 | 1.021 | 2.585 | 1.038 |
| | 48 | 1.868 | 0.910 | 2.478 | 1.129 | 2.370 | 1.084 | **1.673** | **0.854** | 2.304 | 1.043 | 2.371 | 1.007 | 3.386 | 1.186 | 2.130 | 1.024 | 2.505 | 1.041 | 3.024 | 1.145 |
| | 60 | 1.928 | 0.933 | 2.126 | 1.035 | 2.578 | 1.150 | **1.529** | **0.862** | 2.354 | 1.046 | 2.694 | 1.112 | 3.658 | 1.268 | 2.368 | 1.096 | 2.742 | 1.122 | 2.761 | 1.114 |
| | Avg | 1.919 | 0.912 | 2.347 | 1.101 | 2.350 | 1.084 | **1.538** | **0.841** | 2.345 | 1.037 | 2.526 | 1.060 | 3.394 | 1.215 | 2.169 | 1.041 | 2.596 | 1.069 | 2.819 | 1.119 |
| Exchange | 96 | **0.085** | **0.201** | 0.086 | 0.212 | 0.118 | 0.256 | 0.087 | 0.209 | 0.160 | 0.296 | 0.159 | 0.312 | 0.312 | 0.420 | 0.086 | 0.208 | 0.139 | 0.276 | 0.197 | 0.323 |
| | 192 | 0.180 | **0.298** | 0.191 | 0.325 | 0.341 | 0.448 | 0.190 | 0.312 | 0.380 | 0.451 | 0.198 | 0.344 | 0.589 | 0.604 | **0.163** | 0.299 | 0.256 | 0.369 | 0.300 | 0.369 |
| | 336 | **0.307** | **0.398** | 0.309 | 0.415 | 0.727 | 0.672 | 0.377 | 0.446 | 0.659 | 0.615 | 0.323 | 0.441 | 0.976 | 0.817 | 0.311 | 0.424 | 0.426 | 0.464 | 0.509 | 0.524 |
| | 720 | 1.032 | 0.761 | 1.277 | 0.850 | **0.795** | **0.685** | 0.915 | 0.699 | 1.276 | 0.866 | 1.082 | 0.800 | 1.317 | 0.933 | 1.107 | 0.791 | 1.090 | 0.800 | 1.447 | 0.941 |
| | Avg | 0.401 | 0.415 | 0.466 | 0.451 | 0.495 | 0.515 | **0.392** | **0.416** | 0.618 | 0.557 | 0.440 | 0.474 | 0.798 | 0.693 | 0.416 | 0.430 | 0.477 | 0.477 | 0.613 | 0.539 |
| Weather | 96 | **0.143** | **0.198** | 0.159 | 0.228 | 0.156 | 0.212 | 0.152 | 0.199 | 0.163 | 0.223 | 0.170 | 0.235 | 0.147 | 0.211 | 0.176 | 0.237 | 0.238 | 0.314 | 0.249 | 0.329 |
| | 192 | **0.188** | **0.240** | 0.194 | 0.264 | 0.200 | 0.253 | 0.197 | 0.243 | 0.218 | 0.266 | 0.214 | 0.277 | 0.194 | 0.261 | 0.192 | 0.246 | 0.275 | 0.329 | 0.325 | 0.370 |
| | 336 | **0.235** | **0.280** | 0.256 | 0.319 | 0.253 | 0.300 | 0.249 | 0.283 | 0.280 | 0.306 | 0.278 | 0.326 | 0.246 | 0.306 | 0.240 | 0.287 | 0.339 | 0.377 | 0.351 | 0.391 |
| | 720 | **0.310** | **0.336** | 0.315 | 0.364 | 0.314 | 0.346 | 0.320 | **0.335** | 0.349 | 0.356 | 0.318 | 0.363 | 0.322 | 0.363 | 0.316 | 0.352 | 0.389 | 0.409 | 0.415 | 0.426 |
| | Avg | **0.219** | **0.263** | 0.231 | 0.294 | 0.231 | 0.278 | 0.229 | 0.265 | 0.252 | 0.287 | 0.245 | 0.300 | 0.227 | 0.285 | 0.231 | 0.280 | 0.310 | 0.357 | 0.335 | 0.379 |
| Traffic | 96 | 0.394 | 0.274 | 0.397 | 0.279 | 0.377 | 0.266 | **0.367** | **0.251** | 0.603 | 0.328 | 0.461 | 0.290 | 0.489 | 0.276 | 0.410 | 0.282 | 0.576 | 0.359 | 0.597 | 0.371 |
| | 192 | 0.407 | 0.279 | 0.407 | 0.282 | 0.393 | 0.275 | **0.385** | **0.259** | 0.610 | 0.329 | 0.482 | 0.302 | 0.503 | 0.281 | 0.423 | 0.287 | 0.610 | 0.380 | 0.607 | 0.382 |
| | 336 | 0.416 | 0.283 | 0.417 | 0.289 | 0.403 | 0.280 | **0.398** | **0.265** | 0.610 | 0.330 | 0.487 | 0.300 | 0.528 | 0.292 | 0.436 | 0.296 | 0.608 | 0.375 | 0.623 | 0.387 |
| | 720 | 0.453 | 0.305 | 0.455 | 0.311 | 0.441 | 0.300 | **0.434** | **0.287** | 0.632 | 0.352 | 0.527 | 0.310 | 0.593 | 0.326 | 0.466 | 0.315 | 0.621 | 0.375 | 0.639 | 0.395 |
| | Avg | 0.417 | 0.285 | 0.419 | 0.290 | 0.404 | 0.280 | **0.396** | **0.265** | 0.616 | 0.334 | 0.489 | 0.300 | 0.528 | 0.293 | 0.433 | 0.295 | 0.603 | 0.372 | 0.616 | 0.383 |
| Electricity | 96 | **0.130** | 0.226 | 0.140 | 0.235 | 0.133 | 0.232 | **0.130** | **0.222** | 0.181 | 0.281 | 0.162 | 0.272 | 0.198 | 0.292 | 0.140 | 0.237 | 0.186 | 0.302 | 0.196 | 0.313 |
| | 192 | **0.147** | **0.240** | 0.148 | 0.246 | 0.147 | **0.240** | 0.148 | 0.240 | 0.193 | 0.293 | 0.176 | 0.285 | 0.266 | 0.330 | 0.153 | 0.249 | 0.197 | 0.311 | 0.211 | 0.324 |
| | 336 | **0.163** | **0.257** | 0.171 | 0.264 | 0.164 | 0.261 | 0.167 | 0.261 | 0.205 | 0.312 | 0.194 | 0.301 | 0.353 | 0.384 | 0.169 | 0.267 | 0.213 | 0.328 | 0.214 | 0.327 |
| | 720 | **0.198** | **0.290** | 0.209 | 0.297 | 0.203 | 0.291 | 0.202 | 0.291 | 0.222 | 0.320 | 0.222 | 0.327 | 0.400 | 0.416 | 0.203 | 0.301 | 0.233 | 0.344 | 0.236 | 0.342 |
| | Avg | **0.159** | **0.253** | 0.167 | 0.261 | 0.162 | 0.256 | 0.161 | **0.253** | 0.200 | 0.301 | 0.188 | 0.296 | 0.304 | 0.355 | 0.166 | 0.263 | 0.207 | 0.321 | 0.214 | 0.326 |
| 1st count | | **61** | | 0 | | 4 | | 31 | | 0 | | 0 | | 0 | | 1 | | 0 | | 0 | |

∗ We replace the input length L=512 in TimesNet and MICN for a fair comparison. Other experimental results are taken from the PatchTST.

Table 17: All results for multivariate short-input length prediction. The input sequence length is set to 36 for the ILI dataset and 96 for the others. The 1st count indicates the numbers of best performance.

| Methods | WinNet (Ours) | | RLinear (2023a) | | RMLP (2023a) | | PatchTST (2023) | | TimesNet (2023) | | MICN (2023) | | Crossformer (2023) | | DLinear (2023) | | FEDformer (2022) | | Autoformer (2021) | |
|---|---|---|---|---|---|---|---|---|---|---|---|---|---|---|---|---|---|---|---|---|
| Metric | MSE | MAE | MSE | MAE | MSE | MAE | MSE | MAE | MSE | MAE | MSE | MAE | MSE | MAE | MSE | MAE | MSE | MAE | MSE | MAE |
| ETTm1 96 | **0.316** | **0.356** | 0.338 | 0.373 | 0.342 | 0.377 | 0.320 | 0.359 | 0.316 | 0.362 | 0.338 | 0.375 | 0.355 | 0.391 | 0.345 | 0.372 | 0.379 | 0.419 | 0.505 | 0.475 |
| ETTm1 192 | 0.368 | **0.383** | 0.379 | 0.395 | 0.382 | 0.399 | 0.364 | 0.381 | 0.363 | 0.390 | 0.374 | 0.387 | 0.416 | 0.433 | 0.380 | 0.389 | 0.426 | 0.441 | 0.553 | 0.496 |
| ETTm1 336 | 0.398 | 0.402 | 0.403 | 0.407 | 0.404 | 0.416 | 0.391 | 0.401 | 0.408 | 0.426 | 0.410 | 0.411 | 0.486 | 0.479 | 0.413 | 0.413 | 0.445 | 0.459 | 0.621 | 0.537 |
| ETTm1 720 | **0.443** | **0.431** | 0.461 | 0.440 | 0.473 | 0.462 | 0.455 | 0.439 | 0.481 | 0.476 | 0.478 | 0.450 | 0.624 | 0.570 | 0.474 | 0.453 | 0.543 | 0.490 | 0.671 | 0.561 |
| ETTm1 Avg | **0.381** | **0.385** | 0.395 | 0.404 | 0.400 | 0.414 | 0.382 | 0.395 | 0.392 | 0.413 | 0.400 | 0.405 | 0.470 | 0.468 | 0.403 | 0.406 | 0.448 | 0.452 | 0.587 | 0.517 |
| ETTm2 96 | **0.171** | **0.252** | 0.188 | 0.281 | 0.185 | 0.273 | 0.181 | 0.265 | 0.179 | 0.275 | 0.187 | 0.267 | 0.356 | 0.388 | 0.193 | 0.292 | 0.203 | 0.287 | 0.255 | 0.339 |
| ETTm2 192 | **0.240** | **0.300** | 0.288 | 0.361 | 0.252 | 0.321 | 0.247 | 0.307 | 0.307 | 0.376 | 0.249 | 0.309 | 0.422 | 0.440 | 0.284 | 0.362 | 0.269 | 0.328 | 0.281 | 0.340 |
| ETTm2 336 | **0.295** | **0.336** | 0.326 | 0.368 | 0.402 | 0.416 | 0.308 | 0.348 | 0.325 | 0.388 | 0.321 | 0.351 | 0.507 | 0.494 | 0.369 | 0.427 | 0.325 | 0.366 | 0.339 | 0.372 |
| ETTm2 720 | **0.399** | **0.395** | 0.445 | 0.452 | 0.479 | 0.470 | 0.406 | 0.401 | 0.502 | 0.490 | 0.408 | 0.403 | 0.598 | 0.552 | 0.554 | 0.522 | 0.421 | 0.415 | 0.433 | 0.432 |
| ETTm2 Avg | **0.276** | **0.320** | 0.312 | 0.366 | 0.330 | 0.370 | 0.285 | 0.330 | 0.328 | 0.382 | 0.291 | 0.332 | 0.470 | 0.468 | 0.350 | 0.400 | 0.304 | 0.349 | 0.327 | 0.370 |
| ETTh1 96 | 0.379 | **0.388** | 0.398 | 0.416 | 0.378 | 0.396 | 0.392 | 0.408 | 0.421 | 0.431 | 0.389 | 0.412 | 0.409 | 0.432 | 0.386 | 0.400 | **0.376** | 0.419 | 0.449 | 0.459 |
| ETTh1 192 | **0.432** | **0.418** | 0.450 | 0.445 | 0.433 | 0.432 | 0.450 | 0.433 | 0.474 | 0.487 | 0.442 | 0.442 | 0.458 | 0.459 | 0.437 | 0.432 | **0.420** | 0.448 | 0.500 | 0.482 |
| ETTh1 336 | **0.474** | **0.439** | 0.484 | 0.463 | 0.495 | 0.471 | 0.518 | 0.477 | 0.569 | 0.551 | 0.491 | 0.469 | 0.509 | 0.492 | 0.481 | 0.459 | **0.459** | 0.465 | 0.521 | 0.496 |
| ETTh1 720 | **0.472** | **0.457** | 0.531 | 0.524 | 0.540 | 0.517 | 0.522 | 0.490 | 0.770 | 0.672 | 0.521 | 0.500 | 0.696 | 0.632 | 0.519 | 0.516 | 0.506 | 0.507 | 0.514 | 0.512 |
| ETTh1 Avg | **0.439** | **0.425** | 0.466 | 0.462 | 0.462 | 0.454 | 0.470 | 0.452 | 0.558 | 0.535 | 0.460 | 0.455 | 0.518 | 0.503 | 0.455 | 0.451 | 0.440 | 0.459 | 0.496 | 0.487 |
| ETTh2 96 | **0.289** | **0.337** | 0.300 | 0.358 | 0.314 | 0.374 | 0.297 | 0.347 | 0.299 | 0.364 | 0.340 | 0.374 | 0.402 | 0.425 | 0.333 | 0.387 | 0.358 | 0.397 | 0.346 | 0.388 |
| ETTh2 192 | **0.375** | **0.391** | 0.409 | 0.428 | 0.425 | 0.437 | 0.390 | 0.403 | 0.441 | 0.454 | 0.402 | 0.414 | 0.452 | 0.456 | 0.477 | 0.476 | 0.429 | 0.439 | 0.456 | 0.452 |
| ETTh2 336 | **0.416** | **0.428** | 0.489 | 0.482 | 0.480 | 0.472 | 0.417 | 0.429 | 0.654 | 0.567 | 0.452 | 0.452 | 0.533 | 0.506 | 0.594 | 0.541 | 0.496 | 0.487 | 0.482 | 0.486 |
| ETTh2 720 | **0.423** | **0.445** | 0.641 | 0.566 | 0.839 | 0.645 | 0.432 | 0.448 | 0.956 | 0.716 | 0.462 | 0.468 | 0.577 | 0.557 | 0.831 | 0.657 | 0.463 | 0.474 | 0.515 | 0.511 |
| ETTh2 Avg | **0.375** | **0.400** | 0.460 | 0.459 | 0.515 | 0.482 | 0.384 | 0.406 | 0.587 | 0.525 | 0.414 | 0.427 | 0.491 | 0.486 | 0.558 | 0.515 | 0.436 | 0.449 | 0.449 | 0.459 |
| ILI 24 | 2.445 | 0.963 | 3.044 | 1.224 | 3.130 | 1.200 | **1.743** | **0.814** | 2.684 | 1.112 | 2.317 | 0.934 | 4.721 | 1.524 | 2.398 | 1.040 | 3.228 | 1.260 | 3.483 | 1.287 |
| ILI 36 | 2.465 | 1.008 | 3.042 | 1.178 | 3.568 | 1.291 | **1.579** | **0.804** | 2.667 | 1.068 | 1.972 | 0.920 | 4.148 | 1.379 | 2.646 | 1.088 | 2.679 | 1.080 | 3.103 | 1.148 |
| ILI 48 | 2.296 | 0.961 | 2.974 | 1.174 | 3.529 | 1.290 | **2.199** | **0.897** | 2.558 | 1.052 | 2.238 | 0.940 | 4.023 | 1.354 | 2.614 | 1.086 | 2.622 | 1.078 | 2.669 | 1.085 |
| ILI 60 | 2.348 | 0.977 | 3.184 | 1.232 | 3.706 | 1.338 | **1.813** | **0.868** | 2.747 | 1.110 | 2.027 | 0.928 | 4.114 | 1.369 | 2.804 | 1.146 | 2.857 | 1.157 | 2.770 | 1.125 |
| ILI Avg | 2.388 | 0.977 | 3.061 | 1.202 | 3.483 | 1.280 | **1.833** | **0.845** | 2.664 | 1.085 | 2.138 | 0.930 | 4.251 | 1.406 | 2.615 | 1.090 | 2.846 | 1.143 | 3.006 | 1.161 |
| exchange 96 | **0.082** | **0.198** | 0.083 | 0.210 | 0.093 | 0.229 | **0.082** | **0.198** | 0.099 | 0.240 | 0.107 | 0.234 | 0.253 | 0.364 | 0.088 | 0.218 | 0.148 | 0.278 | 0.197 | 0.323 |
| exchange 192 | 0.173 | **0.294** | **0.158** | **0.294** | 0.184 | 0.317 | 0.173 | 0.295 | 0.198 | 0.354 | 0.226 | 0.344 | 0.482 | 0.517 | 0.176 | 0.315 | 0.271 | 0.380 | 0.300 | 0.369 |
| exchange 336 | 0.327 | 0.412 | **0.292** | 0.414 | 0.385 | 0.457 | 0.333 | 0.415 | 0.302 | 0.415 | 0.367 | 0.448 | 0.908 | 0.748 | 0.313 | 0.427 | 0.460 | 0.500 | 0.509 | 0.524 |
| exchange 720 | 0.911 | 0.722 | 0.824 | 0.684 | 0.922 | 0.723 | 0.880 | 0.700 | **0.738** | **0.662** | 0.964 | 0.746 | 1.414 | 0.975 | 0.839 | 0.695 | 1.195 | 0.841 | 1.447 | 0.941 |
| exchange Avg | 0.373 | 0.406 | 0.339 | **0.401** | 0.396 | 0.432 | 0.367 | 0.402 | **0.334** | 0.425 | 0.416 | 0.443 | 0.764 | 0.651 | 0.354 | 0.413 | 0.518 | 0.499 | 0.613 | 0.539 |
| Weather 96 | 0.164 | 0.223 | 0.198 | 0.262 | 0.179 | 0.234 | 0.177 | **0.218** | **0.161** | 0.229 | 0.172 | 0.220 | 0.162 | 0.231 | 0.196 | 0.255 | 0.217 | 0.296 | 0.266 | 0.336 |
| Weather 192 | **0.213** | 0.268 | 0.237 | 0.295 | 0.217 | 0.268 | 0.225 | **0.259** | 0.220 | 0.281 | 0.219 | 0.261 | 0.211 | 0.281 | 0.237 | 0.296 | 0.276 | 0.336 | 0.307 | 0.367 |
| Weather 336 | **0.271** | 0.313 | 0.285 | 0.337 | 0.265 | 0.306 | 0.275 | **0.296** | 0.278 | 0.331 | 0.280 | 0.306 | 0.270 | 0.328 | 0.283 | 0.335 | 0.339 | 0.380 | 0.359 | 0.395 |
| Weather 720 | 0.354 | 0.370 | 0.346 | 0.385 | 0.341 | 0.372 | 0.351 | 0.346 | **0.311** | 0.356 | 0.365 | 0.359 | 0.352 | 0.382 | 0.345 | 0.381 | 0.403 | 0.428 | 0.419 | 0.428 |
| Weather Avg | 0.250 | 0.293 | 0.267 | 0.320 | 0.251 | 0.295 | 0.257 | **0.279** | 0.242 | 0.299 | 0.259 | 0.286 | 0.248 | 0.305 | 0.265 | 0.316 | 0.308 | 0.360 | 0.337 | 0.381 |
| Traffic 96 | **0.421** | 0.344 | 0.657 | 0.403 | 0.540 | 0.351 | 0.540 | 0.357 | 0.519 | 0.309 | 0.593 | 0.321 | 0.512 | **0.288** | 0.650 | 0.396 | 0.587 | 0.366 | 0.613 | 0.388 |
| Traffic 192 | **0.452** | 0.350 | 0.603 | 0.377 | 0.526 | 0.344 | 0.536 | 0.352 | 0.537 | 0.315 | 0.617 | 0.336 | 0.538 | **0.297** | 0.598 | 0.370 | 0.604 | 0.373 | 0.616 | 0.382 |
| Traffic 336 | **0.475** | 0.362 | 0.610 | 0.380 | 0.539 | 0.347 | 0.547 | 0.355 | 0.534 | **0.313** | 0.629 | 0.336 | 0.569 | 0.315 | 0.605 | 0.373 | 0.621 | 0.383 | 0.622 | 0.337 |
| Traffic 720 | **0.483** | 0.371 | 0.651 | 0.401 | 0.577 | 0.366 | 0.541 | 0.330 | 0.577 | **0.325** | 0.640 | 0.350 | 0.613 | 0.336 | 0.645 | 0.394 | 0.626 | 0.382 | 0.660 | 0.408 |
| Traffic Avg | **0.457** | 0.356 | 0.630 | 0.390 | 0.546 | 0.352 | 0.541 | 0.348 | 0.541 | 0.315 | 0.619 | 0.335 | 0.558 | **0.309** | 0.624 | 0.383 | 0.609 | 0.376 | 0.627 | 0.378 |
| Electricity 96 | 0.167 | **0.259** | 0.199 | 0.286 | 0.184 | 0.271 | 0.208 | 0.297 | **0.164** | 0.269 | 0.168 | 0.272 | 0.224 | 0.310 | 0.197 | 0.282 | 0.193 | 0.308 | 0.201 | 0.317 |
| Electricity 192 | **0.177** | **0.265** | 0.187 | 0.281 | 0.187 | 0.276 | 0.198 | 0.288 | **0.177** | 0.285 | 0.184 | 0.289 | 0.281 | 0.345 | 0.196 | 0.285 | 0.201 | 0.315 | 0.222 | 0.334 |
| Electricity 336 | **0.193** | **0.282** | 0.200 | 0.295 | 0.201 | 0.292 | 0.200 | 0.285 | **0.193** | 0.300 | 0.198 | 0.300 | 0.351 | 0.394 | 0.209 | 0.301 | 0.214 | 0.329 | 0.231 | 0.338 |
| Electricity 720 | 0.232 | **0.317** | 0.236 | 0.328 | 0.236 | 0.324 | 0.241 | 0.318 | **0.212** | 0.321 | 0.220 | 0.320 | 0.426 | 0.439 | 0.245 | 0.333 | 0.246 | 0.355 | 0.254 | 0.361 |
| Electricity Avg | 0.192 | **0.280** | 0.206 | 0.298 | 0.202 | 0.291 | 0.211 | 0.297 | **0.186** | 0.294 | 0.192 | 0.295 | 0.320 | 0.372 | 0.211 | 0.300 | 0.213 | 0.326 | 0.227 | 0.337 |
| 1st count | **51** | | 4 | | 0 | | 19 | | 15 | | 0 | | 3 | | 0 | | 3 | | 0 | |

∗ We replace the input length L=96 in PatchTST and Crossformer for a fair comparison. Other experimental results are taken from the TimesNet and PETformer.

Table 18: All results for univariate long-input length prediction. The input sequence length is set to 104 for the ILI dataset and 336 for the others. The 1st count indicates the numbers of the performance.

| Methods | | WinNet (Ours) | | RLinear (2023a) | | RMLP (2023a) | | PatchTST (2023) | | TimesNet (2023) | | MICN (2023) | | DLinear (2023) | | FEDformer (2022) | | Autoformer (2021) | |
|---|---|---|---|---|---|---|---|---|---|---|---|---|---|---|---|---|---|---|---|
| Metric | | MSE | MAE | MSE | MAE | MSE | MAE | MSE | MAE | MSE | MAE | MSE | MAE | MSE | MAE | MSE | MAE | MSE | MAE |
| ETTm1 | 96 | 0.025 | 0.120 | 0.029 | 0.127 | 0.053 | 0.179 | 0.026 | 0.121 | 0.028 | 0.126 | 0.027 | 0.123 | 0.028 | 0.123 | 0.033 | 0.140 | 0.056 | 0.183 |
| | 192 | 0.038 | 0.148 | 0.043 | 0.154 | 0.052 | 0.174 | 0.039 | 0.150 | 0.048 | 0.167 | 0.043 | 0.154 | 0.045 | 0.156 | 0.058 | 0.186 | 0.081 | 0.216 |
| | 336 | 0.051 | 0.171 | 0.064 | 0.187 | 0.111 | 0.265 | 0.053 | 0.173 | 0.060 | 0.188 | 0.052 | 0.173 | 0.061 | 0.182 | 0.084 | 0.231 | 0.076 | 0.218 |
| | 720 | 0.062 | 0.189 | 0.081 | 0.212 | 0.079 | 0.213 | 0.074 | 0.207 | 0.076 | 0.213 | 0.075 | 0.206 | 0.080 | 0.210 | 0.102 | 0.250 | 0.110 | 0.267 |
| | Avg | 0.044 | 0.157 | 0.054 | 0.170 | 0.073 | 0.207 | 0.048 | 0.162 | 0.053 | 0.173 | 0.049 | 0.164 | 0.053 | 0.167 | 0.069 | 0.201 | 0.080 | 0.221 |
| ETTm2 | 96 | 0.062 | 0.181 | 0.067 | 0.193 | 0.070 | 0.197 | 0.065 | 0.186 | 0.076 | 0.206 | 0.063 | 0.183 | 0.063 | 0.183 | 0.067 | 0.198 | 0.065 | 0.189 |
| | 192 | 0.090 | 0.224 | 0.095 | 0.233 | 0.098 | 0.239 | 0.094 | 0.231 | 0.107 | 0.251 | 0.091 | 0.225 | 0.092 | 0.227 | 0.102 | 0.245 | 0.118 | 0.256 |
| | 336 | 0.116 | 0.258 | 0.122 | 0.266 | 0.159 | 0.313 | 0.120 | 0.265 | 0.135 | 0.284 | 0.121 | 0.265 | 0.119 | 0.261 | 0.130 | 0.279 | 0.154 | 0.305 |
| | 720 | 0.168 | 0.318 | 0.173 | 0.320 | 0.203 | 0.353 | 0.171 | 0.322 | 0.210 | 0.362 | 0.172 | 0.317 | 0.175 | 0.320 | 0.178 | 0.325 | 0.182 | 0.335 |
| | Avg | 0.109 | 0.246 | 0.114 | 0.253 | 0.132 | 0.275 | 0.112 | 0.251 | 0.132 | 0.275 | 0.111 | 0.247 | 0.112 | 0.247 | 0.119 | 0.261 | 0.129 | 0.271 |
| ETTh1 | 96 | 0.052 | 0.176 | 0.059 | 0.183 | 0.078 | 0.221 | 0.055 | 0.179 | 0.062 | 0.195 | 0.059 | 0.190 | 0.056 | 0.180 | 0.079 | 0.215 | 0.071 | 0.206 |
| | 192 | 0.068 | 0.203 | 0.078 | 0.212 | 0.094 | 0.241 | 0.071 | 0.205 | 0.080 | 0.225 | 0.087 | 0.235 | 0.071 | 0.204 | 0.104 | 0.245 | 0.114 | 0.262 |
| | 336 | 0.080 | 0.225 | 0.100 | 0.248 | 0.116 | 0.270 | 0.081 | 0.225 | 0.075 | 0.215 | 0.089 | 0.237 | 0.098 | 0.244 | 0.119 | 0.270 | 0.107 | 0.258 |
| | 720 | 0.079 | 0.225 | 0.181 | 0.351 | 0.205 | 0.375 | 0.087 | 0.232 | 0.079 | 0.225 | 0.176 | 0.343 | 0.189 | 0.359 | 0.142 | 0.299 | 0.126 | 0.283 |
| | Avg | 0.069 | 0.207 | 0.105 | 0.249 | 0.123 | 0.276 | 0.073 | 0.210 | 0.074 | 0.215 | 0.102 | 0.251 | 0.103 | 0.246 | 0.111 | 0.257 | 0.104 | 0.252 |
| ETTh2 | 96 | 0.128 | 0.277 | 0.136 | 0.287 | 0.140 | 0.293 | 0.129 | 0.282 | 0.151 | 0.310 | 0.128 | 0.271 | 0.131 | 0.279 | 0.128 | 0.271 | 0.153 | 0.306 |
| | 192 | 0.168 | 0.324 | 0.178 | 0.333 | 0.191 | 0.349 | 0.168 | 0.328 | 0.179 | 0.337 | 0.175 | 0.328 | 0.176 | 0.329 | 0.185 | 0.330 | 0.204 | 0.351 |
| | 336 | 0.194 | 0.355 | 0.213 | 0.372 | 0.244 | 0.397 | 0.185 | 0.351 | 0.195 | 0.356 | 0.192 | 0.354 | 0.209 | 0.367 | 0.231 | 0.378 | 0.246 | 0.389 |
| | 720 | 0.222 | 0.380 | 0.293 | 0.442 | 0.314 | 0.458 | 0.224 | 0.383 | 0.195 | 0.363 | 0.268 | 0.418 | 0.276 | 0.426 | 0.278 | 0.420 | 0.268 | 0.409 |
| | Avg | 0.178 | 0.334 | 0.205 | 0.358 | 0.222 | 0.374 | 0.176 | 0.336 | 0.180 | 0.341 | 0.190 | 0.342 | 0.198 | 0.350 | 0.205 | 0.349 | 0.217 | 0.363 |
| Weather | 96 | 0.0010 | 0.0237 | 0.0055 | 0.0604 | 0.0028 | 0.0416 | 0.0012 | 0.0256 | 0.0012 | 0.0257 | 0.0059 | 0.0645 | 0.0055 | 0.0617 | 0.0042 | 0.0533 | 0.0034 | 0.0467 |
| | 192 | 0.0012 | 0.0269 | 0.0061 | 0.0656 | 0.0047 | 0.0535 | 0.0013 | 0.0268 | 0.0013 | 0.0281 | 0.0063 | 0.0674 | 0.0061 | 0.0659 | 0.0067 | 0.0669 | 0.0039 | 0.0482 |
| | 336 | 0.0014 | 0.0284 | 0.0065 | 0.0680 | 0.0046 | 0.0530 | 0.0014 | 0.0283 | 0.0015 | 0.0298 | 0.0075 | 0.0748 | 0.0064 | 0.0678 | 0.0024 | 0.0394 | 0.0077 | 0.0633 |
| | 720 | 0.0019 | 0.0333 | 0.0069 | 0.0706 | 0.0043 | 0.0509 | 0.0019 | 0.0324 | 0.0020 | 0.0345 | 0.0059 | 0.0634 | 0.0068 | 0.0706 | 0.0038 | 0.0510 | 0.0103 | 0.0743 |
| | Avg | 0.0013 | 0.0280 | 0.0063 | 0.0662 | 0.0041 | 0.0498 | 0.0014 | 0.0282 | 0.0015 | 0.0295 | 0.0064 | 0.0675 | 0.0062 | 0.0665 | 0.0042 | 0.0526 | 0.0063 | 0.0581 |
| exchange | 96 | 0.098 | 0.232 | 0.125 | 0.271 | 0.181 | 0.332 | 0.162 | 0.314 | 0.156 | 0.293 | 0.150 | 0.328 | 0.111 | 0.262 | 0.369 | 0.475 | 0.418 | 0.516 |
| | 192 | 0.210 | 0.341 | 0.235 | 0.391 | 0.428 | 0.518 | 0.273 | 0.411 | 0.264 | 0.375 | 0.211 | 0.377 | 0.229 | 0.391 | 0.511 | 0.551 | 0.538 | 0.591 |
| | 336 | 0.412 | 0.480 | 0.395 | 0.499 | 0.628 | 0.632 | 0.479 | 0.538 | 0.501 | 0.536 | 0.440 | 0.524 | 0.427 | 0.514 | 0.713 | 0.635 | 0.978 | 0.760 |
| | 720 | 1.218 | 0.836 | 1.324 | 0.914 | 1.060 | 0.850 | 0.912 | 0.770 | 1.411 | 0.937 | 1.337 | 0.943 | 1.500 | 1.009 | 1.307 | 0.888 | 1.223 | 0.857 |
| | Avg | 0.484 | 0.472 | 0.520 | 0.519 | 0.574 | 0.583 | 0.456 | 0.508 | 0.583 | 0.535 | 0.534 | 0.543 | 0.566 | 0.544 | 0.725 | 0.637 | 0.789 | 0.681 |
| ECL | 96 | 0.202 | 0.314 | 0.207 | 0.319 | 0.222 | 0.331 | 0.247 | 0.348 | 0.227 | 0.338 | 0.244 | 0.355 | 0.208 | 0.321 | 0.396 | 0.471 | 0.482 | 0.519 |
| | 192 | 0.239 | 0.341 | 0.239 | 0.340 | 0.261 | 0.362 | 0.315 | 0.389 | 0.286 | 0.382 | 0.295 | 0.395 | 0.238 | 0.342 | 0.376 | 0.460 | 0.574 | 0.568 |
| | 336 | 0.276 | 0.369 | 0.269 | 0.365 | 0.297 | 0.388 | 0.490 | 0.494 | 0.326 | 0.401 | 0.333 | 0.428 | 0.273 | 0.369 | 0.522 | 0.543 | 0.587 | 0.581 |
| | 720 | 0.317 | 0.413 | 0.313 | 0.414 | 0.355 | 0.443 | 0.550 | 0.540 | 0.392 | 0.455 | 0.397 | 0.472 | 0.310 | 0.411 | 0.531 | 0.555 | 0.564 | 0.567 |
| | Avg | 0.258 | 0.359 | 0.257 | 0.359 | 0.283 | 0.381 | 0.400 | 0.442 | 0.307 | 0.394 | 0.317 | 0.412 | 0.257 | 0.360 | 0.456 | 0.507 | 0.551 | 0.558 |
| traffic | 96 | 0.126 | 0.207 | 0.128 | 0.209 | 0.136 | 0.218 | 0.134 | 0.211 | 0.144 | 0.232 | 0.150 | 0.249 | 0.139 | 0.230 | 0.250 | 0.355 | 0.280 | 0.388 |
| | 192 | 0.131 | 0.214 | 0.131 | 0.214 | 0.138 | 0.223 | 0.139 | 0.219 | 0.142 | 0.228 | 0.146 | 0.225 | 0.140 | 0.232 | 0.216 | 0.325 | 0.292 | 0.398 |
| | 336 | 0.129 | 0.217 | 0.131 | 0.217 | 0.141 | 0.233 | 0.138 | 0.221 | 0.139 | 0.229 | 0.152 | 0.239 | 0.142 | 0.236 | 0.331 | 0.428 | 0.241 | 0.354 |
| | 720 | 0.143 | 0.233 | 0.145 | 0.235 | 0.175 | 0.276 | 0.154 | 0.244 | 0.153 | 0.250 | 0.161 | 0.249 | 0.157 | 0.254 | 0.414 | 0.487 | 0.240 | 0.341 |
| | Avg | 0.132 | 0.217 | 0.133 | 0.218 | 0.148 | 0.237 | 0.141 | 0.223 | 0.144 | 0.234 | 0.152 | 0.240 | 0.144 | 0.238 | 0.302 | 0.398 | 0.263 | 0.370 |
| ILI | 24 | 0.619 | 0.557 | 1.694 | 1.135 | 1.020 | 0.878 | 0.845 | 0.651 | 0.698 | 0.661 | 2.356 | 1.209 | 0.742 | 0.661 | 1.024 | 0.880 | 0.928 | 0.834 |
| | 36 | 0.677 | 0.602 | 1.804 | 1.183 | 1.035 | 0.896 | 0.698 | 0.624 | 0.708 | 0.680 | 0.646 | 0.616 | 0.553 | 0.613 | 1.009 | 0.897 | 0.942 | 0.838 |
| | 48 | 0.683 | 0.633 | 2.030 | 1.265 | 1.114 | 0.929 | 0.806 | 0.711 | 0.792 | 0.740 | 0.955 | 0.859 | 0.700 | 0.706 | 1.014 | 0.886 | 1.032 | 0.881 |
| | 60 | 0.683 | 0.663 | 2.156 | 1.308 | 1.209 | 0.977 | 0.827 | 0.753 | 0.911 | 0.814 | 1.162 | 0.970 | 0.863 | 0.801 | 1.384 | 1.025 | 1.657 | 1.171 |
| | Avg | 0.665 | 0.613 | 1.921 | 1.222 | 1.095 | 0.920 | 0.794 | 0.684 | 0.777 | 0.723 | 1.279 | 0.913 | 0.714 | 0.695 | 1.107 | 0.922 | 1.139 | 0.931 |
| 1st Count | | 67 | | 9 | | 0 | | 13 | | 6 | | 0 | | 5 | | 2 | | 0 | |

∗ We replace the input length L=336 in TimesNet and MICN for a fair comparison. Other experimental results are taken from the PatchTST.

