| Method | FLOPs | Parameter | Time | Memory |
|---|---|---|---|---|
| **WinNet** | **273M** | **836.8K** | 0.42s | **12MiB** |
| PatchTST | 51.1G | 18.18M | 0.17s | 85MiB |
| TimesNet | 1620.1G | 226.3M | 0.70s | 878MiB |
| MICN | 5.95G | 19.07M | 0.04s | 88MiB |
| Crossformer | 146.4G | 11.1M | 0.54s | 96MiB |
| DLinear | 333M | 1.04M | **0.01s** | 14MiB |
| FEDformer | 2.35G | 4.76M | 0.56s | 39MiB |
| Autoformer | 2.35G | 3.19M | 0.26s | 32MiB |
| Informer | 1.84G | 3.35M | 0.14s | 33MiB |
| Transformer | 2.17G | 2.95M | 0.03s | 31MiB |

Table 18: Efficiency of our model on the ETTm1 dataset vs. other methods in multivariate prediction. We set the input length to 720 and the prediction length to 720. See relevant computational efficiency with thop, torchsummary and memory_allocated functions.

| Method | FLOPs | Parameter | Time | Memory |
|---|---|---|---|---|
| **WinNet** | **5.96M** | **830.9K** | 28ms | **11MiB** |
| PatchTST | 309M | 8.69M | 26ms | 44MiB |
| TimesNet | 406.2G | 57.28M | 500ms | 239MiB |
| MICN | 5.33G | 18.75M | 25ms | 85MiB |
| Crossformer | 3.46G | 11.09M | 62ms | 57MiB |
| DLinear | 7.26M | 1.04M | **15ms** | 12MiB |
| FEDformer | 1.75G | 3.96M | 450ms | 24MiB |
| Autoformer | 1.75G | 2.39M | 270ms | 27MiB |
| Informer | 1.42G | 2.78M | 150ms | 29MiB |
| Transformer | 1.75G | 2.39M | 45ms | 27MiB |