# OpenReview forum: "WinNet:time series forecasting with a window-enhanced period extracting and interacting"
_ICLR.cc/2024/Conference — Submitted to ICLR 2024_

### Official Review · Reviewer_7BDu · 2023-10-22

**Soundness:** 2 fair
**Presentation:** 2 fair
**Contribution:** 2 fair
**Rating:** 5
**Confidence:** 2

**Summary:**

The paper introduces WinNet, a CNN-based model tailored for long-term time series forecasting. Traditional Transformer-based approaches, despite their advancements, struggle with computational efficiency and capturing the periodicity of time series data. WinNet seeks to address these issues with its unique architecture:

1. **Inter-Intra Period Encoder (I2PE):** Converts a 1D sequence into a 2D tensor, capturing both long and short periodicities.
2. **Two-Dimensional Period Decomposition (TDPD):** Models period-trend and oscillation terms, emphasizing the significance of periodicity in time series data.
3. **Decomposition Correlation Block (DCB):** Exploits the correlations between period-trend and oscillation terms, enhancing prediction capabilities.

A pivotal concept introduced is the "periodic window," derived as the least common multiple of multiple periods obtained via Fourier Frequency Transformation. This enables the model to represent variations of multiple short periods and organizes the sequence into a 2D tensor, wherein each row signifies a short-period trend and each column stands for the long-period trend.

WinNet's innovations result in a simplified structure with a single convolutional layer at its core, which significantly reduces computational complexity. Moreover, this model outperforms various baselines in both univariate and multivariate prediction tasks across multiple domains, as evidenced by experiments conducted on nine benchmark datasets.

**Strengths:**

**Strengths:**

1. **Originality:**
    - *Periodic Window Concept:* The inception of the periodic window, which is ascertained by the least common multiple of various periods using Fourier Frequency Transformation, offers a groundbreaking method to encapsulate the core characteristics of time series data.

2. **Quality:**
    - *Ease of Implementation:* WinNet is notably straightforward to implement, and empirical results demonstrate that it surpasses many other methods in a majority of scenarios.

3. **Efficiency:**
    - *Optimized Parameters and Complexity:* WinNet has fewer parameters and a reduced computational complexity compared to MLP techniques and other prevailing methods.

4. **Clarity:**
    - *Well-crafted Manuscript:* The paper is articulately penned and provides a seamless reading experience, making it easy for readers to follow and comprehend.

**Weaknesses:**

**Weaknesses:**

1. **Inconsistent Mathematical Notation:**
    - The authors' presentation of mathematical symbols lacks consistency. As an example, function names should conventionally be displayed in regular typeface instead of italic. Furthermore, vectors and matrices should be represented in bold. Adhering to proper notation is pivotal for ensuring clarity and averting potential misunderstandings.

2. **Graphics and Formatting Issues:**
    - The image quality in Figure 4 is noticeably poor. Additionally, the authors did not adhere to ICLR's guidelines, as they combined the appendix and main text, which could hamper structured reading and comprehension.

3. **On CNN-based MTS:**
    - Given the inherent characteristics of time series data, it's a general understanding that Transformer-based methods often underperform compared to Linear and CNN approaches in various scenarios. However, one of the advantages of Transformer methods is their ability to capture interrelationships amongst multi-variables. Evaluating CNN-based methods purely on accuracy might not provide a fair comparison. Moreover, several existing methods, such as Seq-VAE, leverage CNN for MTS tasks. A comparison of WinNet with such methods would have added depth to the evaluation.

**Questions:**

1. **Normalization in MTS:**
    - Normalization plays a pivotal role in predicting MTS. What kind of normalization technique has been employed within the WinNet framework?

2. **Handling Multivariate Time Series in WinNet:**
    - How does WinNet approach and manage the relationships between variables in a multivariate time series (MTS)? A more in-depth discussion on this aspect could enhance the paper's clarity.

3. **Hyperparameter Settings in WinNet:**
    - Could you elaborate on how the hyperparameters for WinNet were determined? Moreover, how do these hyperparameter choices influence the final results? Insights on this could help understand the model's sensitivity and robustness.

---

> ### Author Response · Authors · 2023-11-23
> **Response to Reviewer 7BDu**
>
> **Weaknesses**
>
> (1) Inconsistent Mathematical Notation: The authors' presentation of mathematical symbols lacks consistency. As an example, function names should conventionally be displayed in regular typeface instead of italic. Furthermore, vectors and matrices should be represented in bold. Adhering to proper notation is pivotal for ensuring clarity and averting potential misunderstandings.
>
> **Response:** Thank you for your suggestions. We have updated our manuscript about the presentation of mathematical symbols and matrices.
>
> (2) Graphics and Formatting Issues: The image quality in Figure 4 is noticeably poor. Additionally, the authors did not adhere to ICLR's guidelines, as they combined the appendix and main text, which could hamper structured reading and comprehension.
>
> **Response:** Thank you for your suggestions. In this revision, we have combined the main text and appendix into a single PDF to adhere to ICLR’s guidelines.
>
> (3) On CNN-based MTS: ......
>
> **Response:** Thank you for your comments. As shown in Tables 5, 14 and 15, we have compared not only the accuracy with the transformer model, but also the efficiency of the model including training time, inference time and number of parameters. While the Transformer models have the ability to capture the relationships between multiple variables, WinNet uses a channel-independence strategy (following to DLinear [1], RLinear [2]) to eliminate interactions between multiple channels and focuses on capturing seasonality in a single channel, mostly achieving excellent results.
>
> We have searched for Seq-VAE [3] and find that it is a method applied to sentence generation tasks and is not related to time series forecasting tasks. In addition, the method is a combination of VAE and GAN, and neither the network model nor the CNN is relevant. Would you please provide the above paper in the reference format that can help us find it precisely?
>
> In the original version, we have conducted comparisons with the CNN methods TimesNet [4] and MICN [5], and provided a comprehensive comparison against the MLP methods. In this revision, we have also added a comparison with RLinear and RMLP, as shown in Tables 1 and 2. It is believed that the performance of WinNet can be validated by the comparisons mentioned above.
>
> **Questions**
>
> (1) Normalization in MTS: Normalization plays a pivotal role in predicting MTS. What kind of normalization technique has been employed within the WinNet framework?
>
> **Response:** We adopt the regularization method of RevIN, which is widely used in the SOTA methods, such as DLinear, PatchTST [6], and RLinear.
>
> (2) Handling Multivariate Time Series in WinNet: ......
>
> **Response:** We adopt a channel-independence strategy that eliminates interactions between multiple variables. This strategy enhances the robustness of the model for multivariate time series with different periodic channels, which demonstrates in the modern methods, including DLinear, PatchTST, and RLinear.
>
> (3) Hyperparameter Settings in WinNet: ......
>
> **Response:** In the WinNet experiments, only three parameters need to be determined, including window_len for MLP, kernel_size for Avgpool2D(·), and kernel_size for Conv2D(·). In the original version, we have supplemented the ablation experiments for the window_len as shown in Appendix Table 9. For the Avgpool2D(·) kernel_size, we combine the periodicity of the sequence to set it as w (window size). Furthermore, we have also conducted the ablation studies for kernel_size of Conv2D(·) and the results are presented in Table 10.
>
> [1] Zeng Ailing, et al. “Are Transformers Effective for Time Series Forecasting? ”. arXiv arXiv:2205.13504v3(2022).
>
> [2] Li Zhe, et al. “Revisiting Long-term Time Series Forecasting: An Investigation on Linear Mapping”. arXiv preprint arXiv:2305.10721 (2023).
>
> [3] Gao Ting, et al. “SeqVAE: Sequence variational autoencoder with policy gradient.” Appl Intell 51, 9030–9037 (2021). https://doi.org/10.1007/s10489-021-02374-7
>
> [4] Wu Haixu, et al. “TimesNet: Temporal 2D-Variation Modeling for General Time Series Analysis”. arXiv [arXiv:2210.02186](https://arxiv.org/abs/2210.02186)(2023).
>
> [5] Wang Huiqiang, et al. “MICN: Multi-Scale Local and Global Context Modeling for Long-term Series Forecasting”. https://openreview.net/pdf?id=zt53IDUR1U
>
> [6] Nie Yuqi, et al. “A Time Series is Worth 64 Words: Long-term Forecasting with Transformers”. arXiv [arXiv:2211.14730](https://arxiv.org/abs/2211.14730)(2023).

---

### Official Review · Reviewer_8wnu · 2023-10-30

**Soundness:** 2 fair
**Presentation:** 2 fair
**Contribution:** 2 fair
**Rating:** 5
**Confidence:** 5

**Summary:**

This paper attempts to use the convolutional neural network (CNN) and proposes WinNet for long-term time series forecasting, which is different from most existing works based on Transformer or MLP. Specifically, WinNet first transforms the 1D time series to 2D tensor according to the predefined periodic windows and then performs period-trend and oscillation decomposition. After that, WinNet captures the correlation between period-trend and oscillation terms, based on which convolution and MLP layers are used to get the final prediction. Nine benchmark datasets are used to evaluate the proposed WinNet compared with some baseline methods.

**Strengths:**

This paper attempts to use the convolutional neural network (CNN) for a better long-term time series prediction, which is still seldomly considered.

**Weaknesses:**

1. The blocks in the architecture of WinNet are not introduced clearly in Section 3. e.g.,

 -The I2PE block: How to train the MLP; How to calculate the periodic window when using two periods approximately as one period (11/12 or 23/24) in Table 1; why use n = w; Should we do padding for X_1D before performing Equation (1); What is the detailed process of I2PE in Equation (1) (I can guess the details but they should be described clearly)?

 -The TDPD block: what is the meaning of w×w in Equation (2), is it the kernel_size for AvgPool2d; What are the two inputs in "According to the equation 2, the two inputs can be decomposed into the period-trend and oscillation terms"?

 -The DCB block: What is the process of CI (channel independence strategy) in Equation (3)?

 -The Series Decoder block: How to get  X_{i}^{row } and X_{w ·⌈i/w⌉−(w−1)+(i mod w)}^{col} from X_{output}^{CI} by inter-period convolution and intra-period convolution; What are the details of these two kinds of convolutions; What is the meaning of {w ·⌈i/w⌉−(w−1)+(i mod w)}?

 -Figure 1: What is the process of CA.

2. The proposal WinNet does not compare with SoTA methods and the performance improvement is not significant.

 -It is said in the Appendix that some experimental results are taken from the PatchTST and PETformer, but there is no comparison with PETformer. There are also other MLP-based models (RLinear and RMLP), which outperform PatchTST on some datasets and should be compared with.

   Li, Zhe, et al. "Revisiting Long-term Time Series Forecasting: An Investigation on Linear Mapping." arXiv preprint arXiv:2305.10721 (2023).

 -WinNet cannot beat existing models in many cases even based on the results shown in the manuscript.

 -Why the results of the exchange dataset are not given in Table 2?

 -The architectures for the Ablation Studies are not described clearly. In addition, how about the results by using only inter-period or intra-period branch in Figure 1?

 -Which kinds of Time and Memory are considered in Table 6?

3. Some claims are not clear, e.g., it is not clear why "The correlation between period-trend and oscillation terms can provide the local periodicity in time series".

4. The code is not available for reproducibility.

**Questions:**

Same to the Weaknesses.

---

> ### Author Response · Authors · 2023-11-23
> **Response to Reviewer 8wnu - Part 1**
>
> **Weaknesses**
>
> (1) The I2PE block: How to train the MLP; How to calculate the periodic window when using two periods approximately as one period (11/12 or 23/24) in Table 1; why use n = w; Should we do padding for X_1D before performing Equation (1); What is the detailed process of I2PE in Equation (1) (I can guess the details but they should be described clearly)?
>
> **Response:** MLP is a linear transformation at the dimension of the sequence length, and we set d_model to 576 (the square of periodic window size). We are referring to the period obtained by Fast Fourier Transformation (FFT), and for the odd number of 11 or 23, we simply add 1 to find the common multiple easily.
>
> In the experiments, the number and size of periodic windows are both configured to be the same for generating the square matrix, reinforcing the Avgpool2D(·) operation in the TDPD block. Specifically, the square matrix can balance feature selection between intra-periodic and inter-periodic features during the Avgpool2D(·) operation, which makes the TDPD block more rational.
>
> We perform a linear transformation in the I2PE block to change the length of the sequence to the square of the periodic window size. Then our model performs the padding and Avgpool2D(·) operations to decompose the 2D tensor into the period-trend and oscillation terms. Therefore, it is unable to perform padding before performing Eq. (1).
>
> (2) The TDPD block: what is the meaning of w×w in Equation (2), is it the kernel_size for AvgPool2d; What are the two inputs in "According to the equation 2, the two inputs can be decomposed into the period-trend and oscillation terms"?
>
> **Response:** Yes, wxw is kernel_size for Avgpool2D(·), and we change it to k to differentiate the window size. After the I2PE block, we get the two inputs (features), inter-period and intra-period, which are fed into the TDPD and DCB blocks, respectively. In the series decoder, we perform direct additive fusion of the two features.
>
> (3) The DCB block: What is the process of CI (channel independence strategy) in Equation (3)?
>
> **Response:** We adopt a channel-independence strategy that eliminates interactions between multiple variables. Specifically, we split both intra-periodic and inter-periodic features at the channel dimension and feed the single-channel matrix into the Concat(·) operation to compose a two-channel matrix.
>
> (4) The Series Decoder block: How to get X_{i}\^{row } and X_{w ·⌈i/w⌉−(w−1)+(i mod w)}\^{col} from X_{output}\^{CI} by inter-period convolution and intra-period convolution; What are the details of these two kinds of convolutions; What is the meaning of {w ·⌈i/w⌉−(w−1)+(i mod w)}?
>
> **Response:** X_{i}\^{row} and X_{w -⌈i/w⌉-(w-1)+(i mod w)}\^{col} are the corresponding positions of the time steps at the one-dimension features of the intra-period and inter-period. From the two-dimension matrix, we fuse the (i,j) time steps in the intra-period with the (j,i) time steps in the inter-period. We consider that the intra-period and inter-period trend characteristics are preserved between the points on the main diagonal of the matrix, since they satisfy a progressive relationship in the rows and columns. By modeling these points as the main temporal sequence points, the fusion of the summation can be centred on the diagonal points to get both intra-periodic and inter-periodic features, more fully fusing the periodic features of the sequences. As shown in Table 11, We have conducted ablation studies to validate different fusion mode, and the results demonstrate that direct additive fusion can achieve better performance. **(page 16)**
>
> The intra-period convolutions extract the correlations between period-trend and oscillation terms along the intra-period. The inter-period convolutions extract the correlations among the period. The former concentrates on intra-periodic variations, while the latter focuses on inter-periodic variation. Furthermore, we use parameter-sharing convolution kernels to combine the features of them. We have provided a description about the formula in Series Decoder in Eq. (4). **(page 6)**
>
> (5) Figure 1: What is the process of CA.
>
> **Response:** In the process of CA, we concatenate the single-channel output after the DCB block at the channel dimension for fusion.

---

> ### Author Response · Authors · 2023-11-23
> **Response to Reviewer 8wnu - Part 2**
>
> (6) The proposal WinNet does not compare with .......
>
> **Response:** Thank you for your comments. In the experimental results, we have compared ICLR2023 SOTA models (PatchTST [1], MICN [2], Crossformer [3], TimesNet [4]) and AAAI2023 SOTA model (DLinear [5]). According to your recommendations, we have also added 2 MLP models (RLinear [6] and RMLP), which can be seen in Tables 1 and 2. According to ICLR's REVIEWER GUIDE, papers published on or after 28 May 2023 are not required to compare their work to that paper, so we do not compare with PETformer. In addition, the authors of PETformer [7] do not release the source code and there are no results from the 512 length setting in the paper. Therefore, we have only compared 336 input lengths with PETformer and the results are listed as follows:
>
> |   Methods   | Methods |   WinNet  |   WinNet  |  RLinear  |  RLinear  |    RMLP   |  RMLP | PETformer | PETformer |
> |:-----------:|:-------:|:---------:|:---------:|:---------:|:---------:|:---------:|:-----:|:---------:|:---------:|
> |    Metric   |  Metric |    MSE    |    MAE    |    MSE    |    MAE    |    MSE    |  MAE  |    MSE    |    MAE    |
> |    ETTm1    |    96   | **0.279** |   0.333   |   0.301   |   0.342   |   0.298   | 0.345 |   0.281   | **0.324** |
> |    ETTm1    |   192   | **0.319** |   0.357   |   0.335   |   0.363   |   0.344   | 0.375 |   0.321   | **0.351** |
> |    ETTm1    |   336   | **0.354** |   0.381   |   0.370   |   0.383   |   0.390   | 0.410 |   0.356   | **0.372** |
> |    ETTm1    |   720   | **0.409** |   0.414   |   0.425   |   0.414   |   0.445   | 0.441 |   0.416   | **0.407** |
> |    ETTm2    |    96   | **0.159** |   0.248   |   0.164   |   0.253   |   0.174   | 0.259 |   0.160   | **0.245** |
> |    ETTm2    |   192   | **0.214** | **0.289** |   0.219   |   0.290   |   0.236   | 0.303 |   0.220   | **0.289** |
> |    ETTm2    |   336   | **0.266** |   0.323   |   0.273   |   0.326   |   0.291   | 0.338 |   0.271   | **0.320** |
> |    ETTm2    |   720   | **0.355** | **0.377** |   0.366   |   0.385   |   0.371   | 0.391 |   0.357   |   0.379   |
> |    ETTh1    |    96   |   0.366   |   0.391   |   0.366   |   0.391   |   0.390   | 0.410 | **0.358** | **0.381** |
> |    ETTh1    |   192   |   0.401   |   0.411   |   0.404   |   0.412   |   0.430   | 0.432 | **0.397** | **0.404** |
> |    ETTh1    |   336   |   0.424   |   0.426   |   0.420   |   0.423   |   0.431   | 0.441 | **0.419** | **0.417** |
> |    ETTh1    |   720   | **0.431** | **0.450** |   0.442   |   0.456   |   0.450   | 0.495 |   0.443   |   0.453   |
> |    ETTh2    |    96   |   0.271   |   0.333   | **0.262** | **0.331** |   0.288   | 0.352 |   0.281   |   0.333   |
> |    ETTh2    |   192   |   0.330   | **0.374** | **0.319** | **0.374** |   0.343   | 0.387 |   0.345   |   0.376   |
> |    ETTh2    |   336   |   0.362   |   0.400   | **0.325** |   0.386   |   0.353   | 0.402 |   0.336   | **0.380** |
> |    ETTh2    |   720   |   0.393   |   0.433   | **0.372** | **0.421** |   0.410   | 0.440 |   0.385   |   0.422   |
> |   Weather   |    96   | **0.145** |   0.200   |   0.175   |   0.225   |   0.149   | 0.202 |   0.150   | **0.190** |
> |   Weather   |   192   | **0.188** |   0.246   |   0.218   |   0.260   |   0.194   | 0.242 |   0.194   | **0.232** |
> |   Weather   |   336   | **0.239** |   0.282   |   0.265   |   0.294   |   0.243   | 0.282 |   0.246   | **0.273** |
> |   Weather   |   720   |   0.318   |   0.341   |   0.329   |   0.339   | **0.316** | 0.333 |   0.320   | **0.326** |
> | Electricity |    96   |   0.131   |   0.227   |   0.140   |   0.235   | **0.129** | 0.224 |   0.131   | **0.223** |
> | Electricity |   192   |   0.148   |   0.241   |   0.154   |   0.248   | **0.147** | 0.240 | **0.147** | **0.238** |
> | Electricity |   336   |   0.164   |   0.258   |   0.171   |   0.264   |   0.164   | 0.257 | **0.163** | **0.255** |
> | Electricity |   720   | **0.201** |   0.293   |   0.209   |   0.297   |   0.203   | 0.291 |   0.203   | **0.289** |
>
> [1] Nie Yuqi, et al. “A Time Series is Worth 64 Words: Long-term Forecasting with Transformers”. arXiv arXiv:2211.14730(2023).
>
> [2] Wang Huiqiang, et al. “MICN: Multi-Scale Local and Global Context Modeling for Long-term Series Forecasting”. https://openreview.net/pdf?id=zt53IDUR1U
>
> [3] Zhang Yunhao & Yan Junchi. “Crossformer: Transformer Utilizing Cross-Dimension Dependency for Multivariate Time Series Forecasting.”
>
> [4] Wu Haixu, et al. “TimesNet: Temporal 2D-Variation Modeling for General Time Series Analysis”. arXiv arXiv:2210.02186(2023).
>
> [5] Zeng Ailing, et al. “Are Transformers Effective for Time Series Forecasting? ”. arXiv arXiv:2205.13504v3(2022).
>
> [6] Li Zhe, et al. “Revisiting Long-term Time Series Forecasting: An Investigation on Linear Mapping”. arXiv preprint arXiv:2305.10721 (2023).
>
> [7] Lin Shengsheng, et al. “PETformer: Long-term Time Series Forecasting via Placeholder-enhanced Transformer”. arXiv arXiv:2308.04791v2(2023).

---

> ### Author Response · Authors · 2023-11-23
> **Response to Reviewer 8wnu - Part 3**
>
> (7) WinNet cannot beat existing models in many cases even based on the results shown in the manuscript.
>
> **Response:** Thank you for your insightful comments. As shown in Tables 16, 17 and 18, the proposed WinNet achieves superior performance with longer input lengths (e.g., 336 and 512), while it obtains inferior results with shorter input lengths (e.g., 96). The possible reasons can be explained as follows. In general, the periodicity of the data can be easily extracted as the input lengths become longer. Specifically, we decompose the period-trend and oscillation terms for both intra-periodic and inter-periodic features by Avgpool2D(·) operation. However, the input length of 96 is not sufficient for this purpose. Therefore, WinNet achieves inferior performance for some datasets with 96 input lengths.
>
> (8) Why the results of the exchange dataset are not given in Table 2?
>
> **Response:** We have added experimental results about the exchange dataset in Table 2 **(now is 1)**.
>
> (9) The architectures for the Ablation Studies are not described clearly. In addition, how about the results by using only inter-period or intra-period branch in Figure 1?
>
> **Response:** We have updated the specific details of the ablation experiments in the Appendix A.2 **Model architecture**. Moreover, we have also conducted the ablation studies about only intra-period or inter-period branch. The experimental results of ablation studies show that the method using both intra-period and inter-period achieves the best results. The experimental results can be seen in the Appendix Table 12.
>
> (10) Which kinds of Time and Memory are considered in Table 6?
>
> **Response:** The times in Table 6 (now is 5) are the training times calculated for one iteration, and the memory overhead is obtained by the torch.cuda.memory_allocated function. We have also added the inference times, which presents in Tables 5, 14 and 15. These experimental results demonstrate the efficiency of our model against other SOTA models in univariate prediction, as well as multivariate prediction tasks with different number of channels.
>
> (11) Some claims are not clear, e.g., it is not clear why "The correlation between period-trend and oscillation terms can provide the local periodicity in time series".
>
> **Response:** As for long-term time series forecasting tasks, we conclude that there is an extremely strong lag correlation between the trend and seasonal terms, as shown in Figure 1 (page 2), and a network model should be designed to extract the correlations, rather than modelling these two terms separately, such as DLinear and MICN.
>
> In this work, we concatenate the period-trend and oscillation terms in the DCB block at the channel dimension to form a two-channel matrix and then convolve it into a one-channel matrix. The CNN kernel can exactly extract the variation of the two terms within adjacent periods, and the learned parameters are able to perform a proportional aggregation of them, rather than simply adding. Moreover, the aggregation can extract the lag correlation between the two terms. Therefore, a certain local periodicity can be extracted by a single convolution process.
>
> (12) The code is not available for reproducibility.
>
> **Response:** We will release the source code after the manuscript is accepted.

---

> > ### Comment · Reviewer_8wnu · 2023-12-05
> >
> > Thank authors for the responses. I updated my rating based on the responses and other reviewers' comments especially regarding the presentation.

---

### Official Review · Reviewer_jnhX · 2023-10-31

**Soundness:** 2 fair
**Presentation:** 2 fair
**Contribution:** 3 good
**Rating:** 5
**Confidence:** 3

**Summary:**

In this paper, the authors aim to solve the problems of high computational costs and the missing periodic data capture in forecasting models. To achieve the goal, the authors designed a CNN-based model, WINNET, with one convolutional layer as the backbone. The model includes four parts, Inter-Intra Period Encoder (I2PE), Two-Dimensional Period Decomposition (TDPD), Decomposition Correlation Block (DCB) and Series Decoder. Specifically, I2PE transforms the input 1D sequence into 2D tensor with inter-period and intra-period. TDPD is to obtain the period-trend and oscillation terms. DCB is to study the correlation between the period-trend and oscillation terms. And finally, through Series Decoder, the final prediction results are obtained.
In the experiment, the authors evaluate the performance over the real-world datasets both small and large datasets, comparing to several baselines.

**Strengths:**

Originality:
In this paper, the backbone of the model is mainly a convolution layer, which greatly reduces the computational complexity and improve the efficiency. It is novel and interesting for simplify model structure for time series forecasting tasks. And this try also shows that simple model framework could also effectively perform time series forecasting tasks.
Quality:
From the perspective of quality, it is high. The authors design a new model and demonstrate its effectiveness through detailed explanations. And the data analysis is thorough, well-executed, and adequately supports the conclusions drawn.
Clarity:
In this paper, the introduction provides a clear overview of the research topic and objectives, and the body sections are logically organized. And the language used in this paper is clear and easy to understand. Besides, some key concepts are well explained.
Significance:
The work in this paper is of great significance. Firstly, WINNET outperforms other forecasting models. And then, WINNET harvests the high computational efficiency for other forecasting models and make full use of the correlation between period trend and oscillation.

**Weaknesses:**

(1) In figure 1, through I2PE block, you can get inter-period and intra-period features. The inter-period features represent the long-period features and the intra-period features represent the short-period features. However, in figure 1, you wrote that the short-period features are inter-period features, and the long-period features are intra-period features.
(2) In section 3.1, the framework of I2PE is needed.
(3) In section 3.4, the framework of series decoder is needed.
(4) In the experimental part, I noticed that for some data sets, WINNET's performance is not the best, not even the second best. Some explanation is needed.
(5) Please pay attention to typography issues, such as the size of Table 3. Table 4.

**Questions:**

(1) In section 3.1, you set the number of periodic windows is the same as the periodic window size. What is the reason for this setting?
(2) In the setting of prediction length, the shortest output length is 24. Why not setting the prediction length to the typical prediction length 12?
(3) From figure 4, we can see that in some cases, the performances of other baselines are better than WINNET, which cannot support the conclusion that WINNET outperforms other baselines. Does WINNET outperform only under some certain T settings?

---

> ### Author Response · Authors · 2023-11-23
> **Response to Reviewer jnhX**
>
> **Weaknesses**
>
> (1) In figure 1, through I2PE block, you can get inter-period and intra-period features. The inter-period features represent the long-period features and the intra-period features represent the short-period features. However, in figure 1, you wrote that the short-period features are inter-period features, and the long-period features are intra-period features.
>
> **Response:** Thank you for your correction. We have updated the descriptions in the Figure 1 (now is 2) accordingly. **(page 4)**
>
> (2) In section 3.1, the framework of I2PE is needed.
>
> ####
>
> **Response:** Thank you for your suggestions. We have updated our manuscript and shown the detailed description about the I2PE framework in Eq. (1). **(page 4)**
>
> (3) In section 3.4, the framework of series decoder is needed.
>
> ####
>
> **Response:** We have updated our manuscript and shown the detailed description about the series decoder framework in Eq. (4). **(page 6)**
>
> (4) In the experimental part, I noticed that for some datasets, WINNET's performance is not the best, not even the second best. Some explanation is needed.
>
> **Response:** Thank you for your insightful comments. As shown in Tables 16, 17 and 18, the proposed WinNet achieves superior performance with longer input lengths (e.g., 336 and 512), while it obtains inferior results with shorter input lengths (e.g., 96). The possible reasons can be explained as follows. In general, the periodicity of the data can be easily extracted as the input lengths become longer. Specifically, we decompose the period-trend and oscillation terms for both intra-periodic and inter-periodic features by the Avgpool2D(·) operation. However, the input length of 96 is not sufficient for this purpose. Therefore, WinNet achieves inferior performance for some datasets with 96 input lengths.
>
> (5) Please pay attention to typography issues, such as the size of Table 3. Table 4.
>
> **Response:** We have updated our manuscript to unify the form of the tables.
>
> ####
>
> **Questions**
>
> (1) In section 3.1, you set the number of periodic windows is the same as the periodic window size. What is the reason for this setting?
>
> **Response:** The number and size of periodic windows are both configured to be the same for generating the square matrix, reinforcing the Avgpool2D(·) operation in the TDPD module. Specifically, the square matrix can balance feature selection between intra-periodic and inter-periodic features during the Avgpool2D(·) operation. This approach avoids the over-extraction of periodicity from either feature.
>
> (2) In the setting of prediction length, the shortest output length is 24. Why not setting the prediction length to the typical prediction length 12?
>
> **Response:** In general, the output lengths of time series forecasting tasks are configured as {96, 192, 336, 720}, except for the ILI dataset. For the ILI dataset, we follow the SOTA setting and set it to {24, 36, 48, 60}. In accordance with your suggestion, we have conducted an experiment with the output length of 12 and obtained the second-best results, as shown below:
>
> |      | WinNet | WinNet |  PatchTST |  PatchTST | TimesNet | TimesNet |  MICN |  MICN | Crossformer | Crossformer | DLinear | DLinear | FEDformer | FEDformer | Autoformer | Autoformer |
> |------|:------:|:------:|:---------:|:---------:|:--------:|:--------:|:-----:|:-----:|:-----------:|:-----------:|:-------:|:-------:|:---------:|:---------:|:----------:|:----------:|
> | Hori |   MSE  |   MAE  |    MSE    |    MAE    |    MSE   |    MAE   |  MSE  |  MAE  |     MSE     |     MAE     |   MSE   |   MAE   |    MSE    |    MAE    |     MSE    |     MAE    |
> |  12  |  1.626 |  0.803 | **1.375** | **0.733** |   2.347  |   1.094  | 6.084 | 1.925 |    3.227    |    1.191    |  2.556  |  1.181  |   2.051   |   1.017   |    2.686   |    1.144   |
>
> (3) From figure 4, we can see that ......?
>
> **Response:** Thank you for your insightful comments. The proposed WinNet achieves inferior results with shorter input lengths. The WinNet model is more capable of multivariate time series forecasting tasks with longer input lengths (e.g., 336 and 512). The possible reasons can be explained as follows. In general, the periodicity of the sequence can be easily extracted as the input lengths become longer. Specifically, we decompose the period-trend and oscillation terms for intra-periodic and inter-periodic features by the Avgpool2D(·) operation. However, the input length of 96 is not sufficient for this purpose. Therefore, WinNet achieves inferior performance for some datasets with 96 input lengths.
>
> As shown in Appendix Figure 9, we have conducted an experimental validation on the ETT datasets, and the experimental results further support the above analyses. **(page 19)**

---

### Official Review · Reviewer_PKs2 · 2023-11-01

**Soundness:** 2 fair
**Presentation:** 2 fair
**Contribution:** 2 fair
**Rating:** 5
**Confidence:** 4

**Summary:**

This work proposed a simple 2D-CNN framework for time series forecasting tasks, which mainly utilizes the multiscale periodic bias and achieves good forecasting accuracy with computational efficiency.

**Strengths:**

The numerical results indicate a strong performance when compared to TimesNet, and even demonstrate a slight advantage or equality when compared to PatchTST.

**Weaknesses:**

The storyline of this work appears to lack depth and insights. The inter-intra period encoder (I2PE) block, while not identical to the Timesnet architecture, is closely resemble it. Furthermore, I find it challenging to comprehend why the TDPD block and DCB block can be applied identically in both inter- and intra-period signals. My limited understanding is that the intra-period signal is simply the transpose of the inter-period signal, and I'm unsure of how the proposed winNet addresses the parallel implementation of TDPD and DCB in such a scenario. While the CNN model has potential as the author suggests, the novelty and motivation of this work seem weakly supported. A more comprehensive explanation and study of the design would be beneficial.

**Questions:**

As stated in the weakness.

---

> ### Author Response · Authors · 2023-11-23
> **Response to Reviewer PKs2**
>
> **Weaknesses**
>
> (1) The storyline of this work appears to lack depth and insights. The inter-intra period encoder (I2PE) block, while not identical to the TimesNet architecture, is closely resemble it.
>
> **Response:** Thank you for your insightful comments. In fact, the design of the I2PE module is inspired by TimesNet [1]. However, compared to TimesNet, the proposed I2PE module is unique in the following ways:
>
> 1) We simplify the overheads of the model by choosing a periodic window to cover multiple periods of variation based on the top-k period values.
>
> 2) We introduce both intra-periodic and inter-periodic features to perform the TDPD and DCB blocks to extract the changes between and within periods, respectively.
>
> In contrast, TimesNet uses multiple convolutional kernels to extract features from the top-k periods, which leads to high experimental overheads.
>
> (2) Furthermore, I find it challenging to comprehend why the TDPD block and DCB block can be applied identically in both inter- and intra-period signals. My limited understanding is that the intra-period signal is simply the transpose of the inter-period signal, and I'm unsure of how the proposed winNet addresses the parallel implementation of TDPD and DCB in such a scenario. While the CNN model has potential as the author suggests, the novelty and motivation of this work seem weakly supported. A more comprehensive explanation and study of the design would be beneficial.
>
> **Response:** As the reviewer’s understanding, the inter-period feature is transposed by the intra-period feature. In the experiments, the number and size of periodic windows are both configured to be the same for generating the square matrix, reinforcing the Avgpool2D(·) operation in the TDPD module. Specifically, the square matrix can balance feature selection between intra-periodic and inter-periodic features during the Avgpool2D(·) operation, which makes the TDPD block more rational.
>
> In time series forecasting tasks, we empirically find that there is an extremely strong lag correlation between the trend and seasonal terms, and a network model should be designed to extract the correlations rather than modelling the two terms separately (such as DLinear [2] and MICN [3]). The core idea of our work is to capture correlations between the trend and seasonal terms obtained by a decomposition process. To this end, both the TDPD and DCB blocks are designed to implement the idea, which achieves SOTA results with longer input lengths. We have updated the analysis of lag correlation for time series data as shown in Figure 1. **(page 2)**
>
> [1] Wu Haixu, et al. “TimesNet: Temporal 2D-Variation Modeling for General Time Series Analysis”. arXiv [arXiv:2210.02186](https://arxiv.org/abs/2210.02186)(2023).
>
> [2] Zeng Ailing, et al. “Are Transformers Effective for Time Series Forecasting? ”. arXiv arXiv:2205.13504v3(2022).
>
> [3] Wang Huiqiang, et al. “MICN: Multi-Scale Local and Global Context Modeling for Long-term Series Forecasting”. https://openreview.net/pdf?id=zt53IDUR1U

---

### Meta-Review · Area_Chair_TH5Z · 2023-12-06

**Metareview:**

The paper proposes a CNN-based time series forecasting model. The reviewers unanimously found that this paper lacks novelty and valuable insights for the community. The authors have attempted to provide counter-arguments, addressing some of the points. However, the novelty of the work remains limited. There are many existing time series models and it's not clear when this method is expected to outperform them, and what characteristics of the data it leverages. Simply stating that it "captures the long and short periodicity of time series" is entirely too vague. To warrant acceptance, the paper would have to bring considerably more novelty in terms of the method or some generalizable insights concerning time series modeling, neither of which is the case.

**Justification For Why Not Higher Score:**

All the reviewers have chosen to borderline reject this paper, so it will be rejected. There are no grounds to overturn reviewer consensus.

**Justification For Why Not Lower Score:**

N/A

---

### Decision · Program_Chairs · 2024-01-16

Reject